# `Model-GLUE`: Democratized LLM Scaling for A Large Model Zoo in the Wild

Xinyu Zhao[*1], Guoheng Sun[*2], Ruisi Cai[*3], Yukun Zhou[*4], Pingzhi Li[*1], Peihao Wang[*3]
Bowen Tan[5], Yexiao He[2], Li Chen[6], Yi Liang[6], Beidi Chen[5], Binhang Yuan[4]
Hongyi Wang[†7], Ang Li[†2], Zhangyang Wang[†3], Tianlong Chen[†1]

[1]UNC CH  [2]UMD  [3]UT Austin  [4]HKUST  [5]CMU  [6]Google  [7]Rutgers University

{xinyu,pingzhi,tianlong}@cs.unc.edu, {ghsun,yexiaohe,angliece}@umd.edu
{ruisi.cai,peihaowang,atlaswang}@utexas.edu
yzhoufw@connect.ust.hk, {btan2,beidic}@andrew.cmu.edu
li.lizliz.chen@gmail.com, yiliang@google.com
biyuan@ust.hk, hongyi.wang.001@rutgers.edu
[*]Equal Contribution [†]Equal Supervision

## Abstract

As Large Language Models (LLMs) excel across tasks and specialized domains, scaling LLMs based on existing models has gained significant attention, which is challenged by potential performance drop when combining disparate models. Various techniques have been proposed to aggregate pre-trained LLMs, including model merging, Mixture-of-Experts, and stacking. Despite their merits, a comprehensive comparison and synergistic application of them to a diverse model zoo is yet to be adequately addressed. In light of this research gap, this paper introduces `Model-GLUE`, a holistic LLM scaling guideline. First, our work starts with a benchmarking of existing LLM scaling techniques, especially selective merging, and variants of mixture. Utilizing the insights from the benchmark results, we formulate a strategy for the selection and aggregation of a heterogeneous model zoo characterizing different architectures and initialization. Our methodology involves clustering mergeable models, selecting a merging strategy, and integrating model clusters through model-level mixture. Finally, evidenced by our experiments on a diverse Llama-2-based model zoo, `Model-GLUE` shows an average performance enhancement of 5.61%, achieved without additional training. Codes are available at `https://github.com/Model-GLUE/Model-GLUE`.

## 1  Introduction

Large Language Models (LLMs) have demonstrated unparalleled capability in a diverse array of natural language tasks, encompassing commonsense reasoning, question answering, and specialized domains such as mathematics and programming [39, 43, 52]. The effectiveness of LLMs is based on the scaling law, which posits that proportionally increasing model and training data size leads to enhanced model performance [27]. Nevertheless, the computation overhead and data requirement surge as LLM continues to scale. With the widespread of open-sourced general or specialized LLMs, aggregating existing models to construct a more versatile LLM emerges as an economical alternative to training a larger LLM from scratch [13, 16, 54]. This not only mitigates the computation cost but also leverages the collective advancements of previous efforts in building LLMs.

Within different methods to combine existing LLMs, a major class is merging [2, 4, 22, 24, 35, 59, 63, 64]. Model merging combines multiple models into a single one of the same size through weight-space transformation. Wortsman et al. [59] first propose to merge a few fine-tuned models as a training trick for the flat loss-landscape, and Ilharco et al. [22] extends it to multi-task scenario, both of which employ the simple averaging. Other works propose more complicated merging methods,

leveraging weight sparsity [63, 64] and non-uniform coefficient [4, 35]. However, they assume that all candidate models are "useful" when merging. While this may hold for small-sized designed model collections, it may not be the case in real-world scenarios given a large and divergent model zoo. How to ensure the benefits of merging different model zoo sizes and similarities, and exclude "harmful" candidates, remains underexplored.

Since merging is limited to the same model structures and initial weights, another alternative is Mixture-of-Experts (MoE) [16]. MoE is a conditional computation architecture that activates only a subset of model parameters for each specific input example [47]. MoE LLMs have already demonstrated performance and computational efficiency advantages over their dense counterparts [15, 25, 30, 68]. In particular, we use a broader term "mixture" to denote the aggregation of existing expert LLMs according to the MoE paradigm, which has been successfully implemented in some recent practices [50, 54, 55]. However, these implementations neglect the inherent flexibility of MoE to integrate different expert models, especially those groups that do not work with merging. Also, the difference and possible synergy between merging and mixing have not been thoroughly investigated. Based on the above challenges, our primary research question is formulated as:

*(Q) Is it feasible to establish a benchmark for selecting and aggregating Large Language Models (LLMs) from an extensive and varied model zoo based on current state-of-the-art model merging and mixture, thereby enhancing the overall competence of the final model?*

To address (Q), we present `Model-GLUE`, a comprehensive benchmark and set of guidelines for LLM scaling. `Model-GLUE` is the first work for LLM scaling encompassing a wide range of model group sizes and variability, with a principal emphasis on the merging and mixture methodologies, and also discussion of model stacking. We first delve into merging scheduling, analyzing strategies for identifying potentially detrimental model candidates and various merging techniques. We then explore a variety of model mixtures as an alternative to merging, covering different mixture granularity, routers architecture, routing input inputs, *etc.* Building upon the insights from model merging and mixture, `Model-GLUE` introduces an efficient and robust LLM scaling recipe for a diverse set of models. It starts with model clustering and progressive merging,

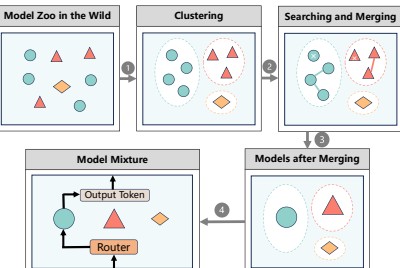

Figure 1: Overview of `Model-GLUE`, composing of (1) Model Clustering based on architecture and weight similarity; (2) Model Filtering and Searching for merging; (3) Model Merging within each cluster; (4) Model Level Mixture of merged models.

and then the mixture of all clusters, thereby integrating similar knowledge from the model zoo while highlighting the respective strengths of each cluster. Our contributions are outlined as follows:

• We conduct a comprehensive benchmarking analysis of LLM merging strategies, beginning with identifying each model's contribution and then followed by filtering out detrimental candidates. Our findings are validated on a range of LLMs, from a few to over a dozen.

• We assess model mixture for four distinct variants: mixture level, router design, router input, and hybrid mixture. We have derived several principles for model mixture and discussed its utility as a solution for scaling models incompatible with merging.

• We introduce a recipe for progressively combining LLM models, `Model-GLUE`, based on findings on merging and mixture benchmarks. It first conducts selective merging and then model mixture, outperforming the best single model on general reasoning, mathematics, and coding tasks.

• Extensive experimental results on Llama-2-based models validate our proposal. For instance, `Model-GLUE` achieves an average increase of $5.61\%$ across chatting, mathematics, and coding benchmarks compared to the best single LLM.

## 2 Related Works

**Model Merging.** Merging methods can be divided into zero-shot merging and merge-then-train approaches. Early zero-shot merging methods are weight averaging and Linear Mode Connectivity [38, 59]. Later popular methods include Task Arithmetic [22] manipulating task vectors, and TIES [63] addressing parameter interference through trimming and conflict resolution. DARE [64] optimizes parameters selectively to enhance merging without extra training. Others focus on geometric properties of weights for merging [49, 24]. Recent Evolutionary Model Merge [4] improves weight configuration and data token pathways during inference. For the merge-then-train approach, Fisher merging [35] uses the Fisher information matrix to weigh model parameters to maximize their joint likelihood. RegMean [26] adapts the linear merging to each linear layer while averaging

embeddings and biases. However, both zero-shot and merge-then-train approaches are less effective for models initialized differently. [2, 23, 53, 62] exploit the permutation symmetry inherent in neural networks on small to large models. To boost merging efficiency, our focus on merging lies in the zero-shot merging of models with the same architecture and initialization.

**Model Mixture.** Mixture-of-Experts (MoE) [47] scales up neural networks by utilizing router networks to activate different parts of the model for different input tokens. Its integration with Large Language Models (LLMs) has gained notable recognition for its exceptional generative capabilities and unparalleled efficiency. Recently, Mixtral [25] demonstrates that the MoE methodology can achieve the performance of dense LLM counterparts while employing significantly fewer active parameters. Model mixture combines a collection of dense LLM models, irrespective of their sizes, into a MoE model. Some studies discover model fusion [54, 55] integrating the outputs of expert models to exploit the unique insights into the data distribution. Recent initiatives include Branch-Train-MiX [50], which starts with a seed-dense LLM and then branches out, facilitating the parallel training of expert models. These trained dense models are subsequently incorporated as experts within MoE layers, with other parameters being averaged. However, this approach is limited to dense models that share identical architectures and sizes. Most recently, UltraFuser [13] introduces a token-level soft gating mechanism that blends model outputs, with a two-stage training strategy.

**Model Stacking.** Model stacking concatenates two models along the depth dimension. In the era of LLM, Wu et al. [60] reuses pre-trained LLaMA layers and resets the output projection to zero in stacking. Kim et al. [28] shows dropping middle layers in stacking yields superior performance. Wang et al. [57] prove that stacking could help recover model-parameter scaling laws with insufficient data. Reddi et al. [42] demonstrated that gradual stacking leads to significant improvements in wall-clock time during the training of few-shot learners. Theoretically, Agarwal et al. [1] proved that model stacking could be interpreted as Nesterov acceleration in network optimization. However, all the aforementioned stacking methods involve no more than two kinds of models and primarily focus on the benefits of training acceleration. In this work, we explore the possibility of stacking two heterogeneous models to combine their capabilities.

**Model Scaling Tools** There have been several tools for model mixture and merging, and for scaling models using existing LLMs. For example, Mergekit is an open-source library designed to facilitate the application of model merging strategies and the construction of MoE [16]. As a representative of unified LLM, Beyonder is a set of mixtures of merged and single LLMs for different tasks[1]. However, there is still a lack of a comprehensive benchmark of the various mixing and merging techniques and practical guidance on how to unify groups of LLMs at different levels of similarity.

## 3 Methodology

### 3.1 Preliminaries

In this study, we consider a collection of $n$ existing Large Language Models (LLMs), denoted as $\{\mathtt{M}_1, \ldots, \mathtt{M}_n\}$, which have been fine-tuned on diverse corpora. Our objective is to outline a systematic approach towards producing one stronger aggregated model across all knowledge domains. Specifically, the unified LLM incorporates single LLMs mainly through merging and mixture.

### 3.2 Model Merging

**The concept of Model Merging** Model merging is integrating multiple models into one unified model in the weight space, compatible with LLMs of the same initialization [16]. Popular merging methods can be divided into two types: ❶ *Merging entire model weights* represented by Model Soup [59] (Linear), SLERP [49], and Model Stock [24]; ❷ *Task-vector based merging* represented by Task Arithmetic [22], TIES [63], and DARE [64]. The former method directly interpolates model weights, while the latter subtracts the pre-trained model from the fine-tuned model to obtain task vectors and utilizes sparsity and consistency of parameters for refined

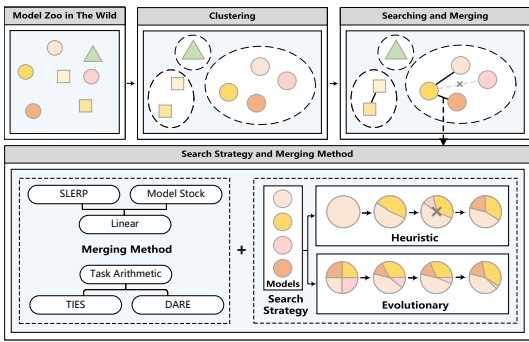

Figure 2: Pipeline for model merging, as well as an overview of merging methods and search strategies.

---

[1] https://huggingface.co/mlabonne/Beyonder-4x7B-v3

merging. The basic Linear interpolation merging is defined as $w_u = \sum_{i=1}^{n} s_i \cdot w_i$, where $w_i$ and $s_i$ are the corresponding model weights and merging coefficient of $M_i \in \{M_1, \ldots M_n\}$.

**Selective Merging Pipeline**   Merging can be easily applied to models with the same architecture, but does not guarantee better results. Therefore, before searching for the merging coefficient, we first pre-process the models by clustering all the models using cosine similarity and then searching for the optimal merging coefficient and method within each cluster. Details are explained in Appendix A.5.

**Heuristic and Evolutionary Strategies**   The heuristic strategy is for searching and filtering potential harmful models for merging. It is based on greedy search, involving three variants: ❶ *Heuristic-Average* retain the candidate if there is an improvement on the proxy dataset in each round of merging. ❷ *Heuristic-Coefficient* builds upon *Heuristic-Average*, by combining the previously merged model with a new candidate using different coefficients in each round. ❸ *Heuristic-Similarity* selects the candidate model with the highest or lowest similarity and conducts a coefficient search to combine it with the previously merged model. Detailed heuristic strategy algorithms can be found in Appendix A.1 Heuristic strategies perform pairwise merging of models, while many methods allow for merging multiple models at once. Therefore, we also consider jointly optimizing all model coefficients using the *Evolutionary Strategy*.

### 3.3   Model Mixture

**The concept of Model Mixture.** Model mixture resembles Mixture-of-Experts(MoE). It scales a LLM with multiple pre-trained LLM experts and further extends beyond traditional token-dependent Feed-Forward-Network (FFN) MoE designs [47]. A mixture model is composed of MoE modules and the rest shared parameters. A MoE module consists of a router $\mathcal{G}(\cdot)$ and $n$ expert networks $\{E_1, \cdots, E_n\}$. $\mathcal{G}(\cdot)$ takes a router input $x_{\mathcal{G}}$ and generate expert assignment for each token input $x$. Then

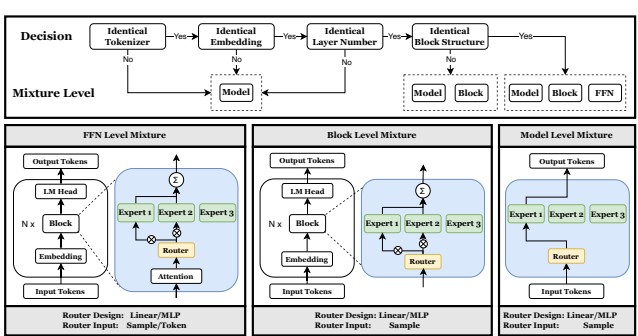

Figure 3: The overview and decision flow of three model mixture levels and their selection philosophy.

MoE outputs a weighted sum of experts' outputs as $\text{MoE}(x, x_{\mathcal{G}}) = \sum_{i=1}^{n} \mathcal{G}(x_{\mathcal{G}})_i \cdot E_i(x)$. We experiment with several variations of Model Mixture, classified as follows:

**Mixture levels.**   Traditional Mixture-of-expert models replace the dense FFN layer at each Transformer block with an MoE module, which is only compatible with LLMs that share the same architecture. Besides this ❶ *FFN level mixture*, we also experiment with two coarse-grained mixtures. ❷ *Block level mixture* create MoE module by aggregating Transformer blocks with the same index from each LLM as experts and add a block-wise router. Block level mixture is applicable to models with different architecture but the same embedding space, layer amounts, and intermediate dimension. ❸ *Model level mixture* take each LLM as an expert and use a router at mixture model input. Model level mixture covers any LLM groups not compatible with FFN and block level mixture. In particular, the model level mixture is similar but not identical to the model ensemble, as the former can be sparse and focus more on efficiency and exploit single LLM expertise, while the latter produces general results by averaging or majority voting overall model outputs. Details can be found in Appendix A.3

**Router design.**   The router network of many MoE studies adheres to a ❶ *linear router* [47]. We experiment with another more complex ❷ *MLP router* to examine whether this router design leads to better performance. It is implemented by two sequential FFN and a ReLU function in between, inspired by [48, 32]. For the routing method, we employ Top-K selection to all routers, which activates the K experts corresponding to the K largest softmaxed router output [47, 48].

**Router input.**   We adopt two types of router input for different levels of model mixture: ❶ Token input for FFN level mixture, where router input is the same as model input; ❷ Sample input for block and model level mixture, where we calculate the average embedding as the sample input $x_{\mathcal{G}} = \sum_{i=1}^{n} x_n$, and route tokens of a sample to the same expert based on sample routing. The sample routing avoids inconsistency in attention operation.

**Hybrid mixture.**   To explore LLM scaling in between model merging and model mixture, we propose the hybrid mixture as an intermediate solution. In a hybrid mixture, the bottom few layers of all single LLMs are merged, and then the rest layers follow any of the mixture level designs.

## 4 Model Merging and Model Mixture for LLMs

### 4.1 Benchmark Datasets and Configs

**Model Zoo.** Table 1 provides an overview of the Model Zoo. For benchmarking model merging and mixture at different sizes of model zoo, we construct 5 groups of Llama-2-based 7B chat LLMs where the number of models $\in [2, 4, 8, 12, 16]$. In addition, to examine the difference in combining models from different domains, we introduce `Which4 (chat)`, consisting of four chat models, as a supplement setting where no single model has a superior advantage in a specific domain.

After comparing the two ways of model scaling, we propose `Model-GLUE` combining selective merging and model mixture, which is tested on the largest model family `Which16`. `Which16` is developed on 12 mergeable Llama-2-based models in `Which12`, which additionally includes four highly performant domain-specific models that cannot be merged: three CodeLlama-based models, two of which are code models and one is a math model, and LLM360/CrystalChat. In particular, LLM360/CrystalChat use different architecture, initialization, and training data from Llama-2-based models, while CodeLlama series, initialized from Llama-2, adopt continuous pretraining rather than fine-tuning as models in `Which12`.

Table 1: All of the models in our model zoos and their performance. For each model zoo, we denote those models that belong to it with a colored star ✧: ✧ for `Which2`, ✧ for `Which4 (Chat)`, ✧ for `Which4 (Domain)`, ✧ for `Which8`, ✧ for `Which12`, and ✧ for `Which16`.

| Model | Model Zoo | ARC | WinoGrande | MMLU | GSM8K | MBPP | HumanEval | Average |
|---|---|---|---|---|---|---|---|---|
| migtissera/Synthia-7B-v1.2 [37, 51, 52] | ✧✧✧✧ | 55.03% | 73.72% | 48.18% | 24.03% | 17.80% | 13.41% | 38.70% |
| neuralmagic/Llama-2-7b-evolcodealpaca [52] | ✧✧✧✧ | 49.57% | 72.45% | 41.70% | 09.02% | 25.60% | 31.71% | 38.34% |
| teknium/OpenHermes-7B [52] | ✧ ✧✧✧ | 56.40% | 73.88% | 47.84% | 09.25% | 22.80% | 19.51% | 38.28% |
| PygmalionAI/pygmalion-2-7b [52] | ✧✧✧✧ | 54.10% | 75.37% | 48.38% | 17.29% | 19.20% | 15.24% | 38.26% |
| meta-llama/Llama-2-7b-chat-hf [52] | ✧✧ ✧✧✧ | 54.10% | 71.27% | 47.28% | 23.05% | 17.00% | 13.41% | 37.68% |
| Severus27/BeingWell_llama2_7b [52] | ✧✧ | 54.95% | 72.30% | 46.19% | 22.29% | 13.40% | 13.41% | 37.09% |
| meta-math/MetaMath-7B-V1.0 [52, 65] | ✧✧✧✧ | 47.35% | 70.24% | 41.58% | 59.06% | 01.40% | 01.22% | 36.81% |
| lmsys/vicuna-7b-v1.5 [66, 52] | ✧✧ ✧✧✧ | 53.75% | 70.56% | 49.78% | 19.11% | 06.00% | 19.51% | 36.45% |
| garage-bAInd/Platypus2-7B [21, 29, 52] | ✧✧✧ | 55.12% | 74.03% | 49.82% | 02.50% | 19.00% | 14.63% | 35.85% |
| GOAT-AI/GOAT-7B-Community [8, 52] | ✧✧ | 49.06% | 72.22% | 49.23% | 09.70% | 05.40% | 09.76% | 32.56% |
| stanford-oval/Llama-2-7b-WikiChat-fused [46, 52] | ✧✧ | 50.94% | 68.59% | 39.13% | 00.00% | 13.80% | 04.27% | 29.45% |
| cognitivecomputations/dolphin-llama2-7b [52] | ✧✧ | 42.66% | 65.35% | 46.52% | 10.69% | 00.80% | 02.44% | 28.08% |
| meta-math/MetaMath-Llemma-7B [7, 65] | ✧ | 46.76% | 64.33% | 46.33% | 62.40% | 42.00% | 31.10% | 48.82% |
| codellama/CodeLlama-7b-Instruct-hf [44] | ✧ | 43.52% | 65.11% | 41.83% | 17.06% | 40.00% | 33.70% | 40.20% |
| ise-uiuc/Magicoder-S-CL-7B [58, 44] | ✧ | 43.77% | 63.38% | 35.94% | 14.33% | 50.20% | 63.41% | 45.17% |
| LLM360/CrystalChat [34] | ✧ | 51.54% | 70.64% | 52.39% | 32.45% | 38.80% | 35.37% | 46.87% |

For merging benchmarks, we experiment with a larger model zoo, namely `Which4`, `Which8`, and `Which12` with models filtered from `Which16`. For model mixture with higher computational cost, we experiment with `Which2` and `Which4`.

**Benchmarks** We assess all models on three categories of benchmarks: (*i*) Commonsense reasoning using ARC [10], WinoGrande [45], and MMLU [20]; (*ii*) Mathematics ability on GSM8K [11]; (*iii*) Coding ability on MBPP [6] and HumanEval [9]. The evaluation scripts are based on lm-eval [2] for commonsense and mathematical reasoning and bigcode-eval [3] for coding datasets. All benchmarks are under the MIT License.

### 4.2 Implementation Details for Merging

**Proxy Dataset.** Since the performance of merging model is not necessarily positive, we need a proxy dataset to determine whether to reject a particular round of merging in the Heuristic Strategy, or to compute the model fitness in the Evolutionary Strategy. (*i*) For MBPP, we select its validation set. (*ii*) For HumanEval, due to the unavailability of a validation set and its smaller size, we select 20% of the JavaScript version of HumanEvalPack [36]. (*iii*) For other tasks, we chose the small-scale datasets released by tinybenchmarks [40] under MIT License.

**Model Zoo and Clustering.** The Merging Bench considers 3 model zoos: `Which4`, `Which8`, and `Which16`. We first cluster the model zoos based on cosine similarity with a threshold of $0.95$. Due to `Which16` contains models that cannot be merged, we choose the mergable family obtained through clustering which is referred to as `Which12`.

**Details of Heuristic Strategy and Evolutionary Strategy.** For *Heuristic Strategy*, to reduce the search space, we only evaluated Linear interpolation and the range of coefficient search is $\{0.1, 0.2...0.9\}$. In *Heuristic-Similarity*, we use the average similarity of all weights as the criterion for selecting models in each round. For *Evolutionary Strategy*, we refer to the setting of Evolutionary Model Merge [4], which utilizes the CMA-ES [19] algorithm implemented by Optuna [3]. In contrast,

---

[2]`https://github.com/EleutherAI/lm-evaluation-harness`

[3]`https://github.com/bigcode-project/bigcode-evaluation-harness`

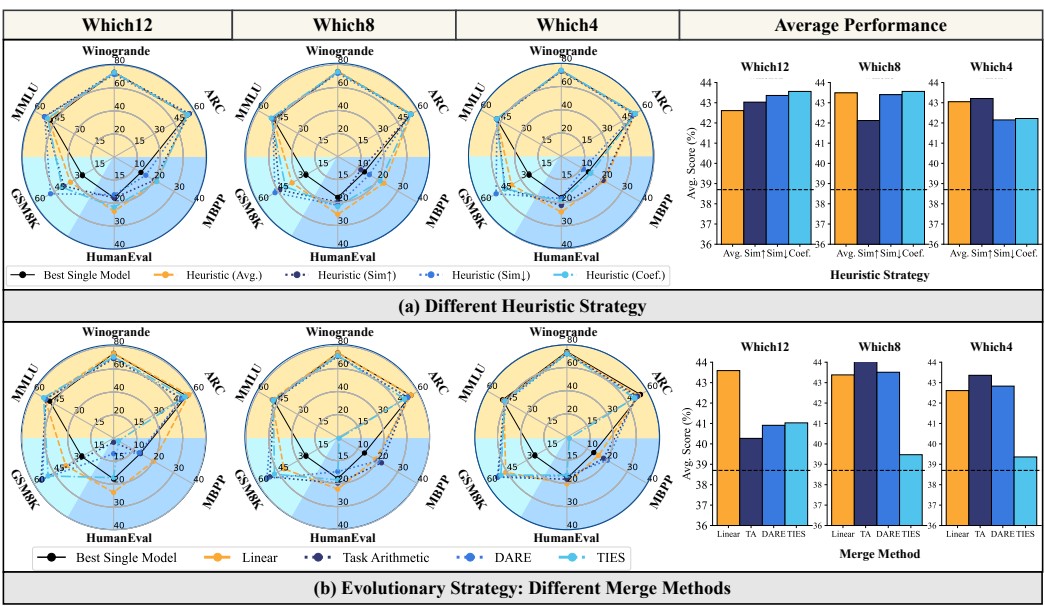

Figure 4: (a) Comparison between different Heuristic Strategies on `Which12`, `Which8`, `Which4`. (b) Comparison of different model merging methods in Evolutionary Strategy.

all parameters are randomly initialized, and the fitness values are defined as the accuracy of the proxy dataset. The optimization was conducted for 200 trials in all scenarios.

## 4.3 Model Merging Benchmark Results

We start our discussion by examining the effectiveness of existing approaches in depth. Despite existing merging methods focus on improving the merging techniques, their effectiveness is usually validated basedt on small-scale model zoos. For instance, Ilharco et al. [22] primarily focuses on the linear interpolation between two fine-tuned models, while Akiba et al. [4] explores merging three.

Current model practitioners typically download pre-trained models, fine-tune them on their own data or with unique techniques for specific downstream tasks, and then upload them back to the public. This practice results in a large number of open-source models being available, yet they remain underutilized by current merging methods. To this end, instead of solely discussing the merging technique, we explore an **orthogonal** question: *Can we scale up the size of model zoo to cover more models, and design an automatic merging technique to benefit from the inclusion?*

**Failure Case of Existing Approaches.** To begin with, we provide a motivating example to show the failure case of the existing approach. We consider the three models, Llama-2-Chat [52], Vicuna [67] and CodeLlama [43], all initialized with the same base model, Llama-2 [52]. We merge Vicuna and CodeLlama with Llama-2-Chat, respectively, and report the evaluation results in Table 14 in Appendix B.2. We evaluate 6 representative merging techniques implemented in *mergekit* [16], including linear interpolation [59], SLERP [49], Model Stock [24], Task Arithmetic [22], DARE [62], and TIES [63]. By merging Llama-2-chat and Vicuna, the merged model achieves better performance compared to any single model, while merging Llama-2-chat and CodeLlama fails to outperform all single models and may even lead to a significant drop in performance, which is also mentioned by Xu et al. [62]. The results indicate the potential severe performance drop when including un-mergeable new model in merging (e.g. CodeLlama). Even if it is obtained from the same pre-trained checkpoint. Such failure case motivates us to design the strategy to automatically select models for merging, and exclude the models that are unable to merge.

In the following paragraphs, we explore several solutions tailored for large-scale model merging. These variations address different resource and speed requirements. The introduction of these methods is organized around answering the following key questions.

**Q1: Does handcrafted rules apply to automated model selection and which one performs best? A: Yes, by a greedy search approach.** In this section, we explore three potential heuristics for model selection and report the results in Figure 4(a). We include the performance of the "best single model" (the model participant before merging that achieves the best averaged performance). We additionally validate the performance of heuristic-based merging technique, which are detailed in Section 3.2. As indicated by the results, the merging technique based on *Heuristic-Coefficient* yields

consistently superior performance when the model zoo is large. For `Which4`, *Heuristic-Average* achieved better performance, while *Heuristic-Coefficient* performed poorly. This is primarily because the domain-specific models in `Which4` exhibit similar performances and are indispensable.

**Q2: How to utilize Evolutionary Strategy for coefficient optimization in model merging?**

We divide the problem into the following sub-questions: (*i*) Which merging method is most compatible with Evolutionary Strategy? (*ii*) Can finer-grained optimization lead to a better merged model? (*iii*) How to efficiently merge in a large model zoo? For (*i*), **A: simpler methods such as Linear and Task Arithmetic are more competitive.** We compared four methods: Linear, Task Arithmetic, DARE, and TIES. As shown in Figure 4(b), Linear merging consistently achieves great results. However, when the parameters to be optimized are small, Task Arithmetic performs slightly better than Linear. Under a fixed computational budget, due to the doubling of parameters to be optimized, DARE and TIES exhibit slightly lower performance compared to other methods. For (*ii*), **A: Yes, but we need a larger computational budget.** We group adjacent $n$ decoder layers together, where they share the same coefficients. The group size $n \in [32, 8, 4, 1]$. When $n = 8$, better results were achieved compared to $n = 32$, as shown in Table 17. However, as we further decreased the group size, the performance slightly declined. This could be attributed to our relatively small budget. For (*iii*), **A: Use Heuristic Strategy to roughly search for coefficients and then fine-tune the coefficients using Evolutionary Strategy.** As shown in Table 18, the combination of the two strategies resulted in better results with fewer trials. For implementation details, please refer to Appendix A.2.

### 4.4 Implementation Details for Mixture

**Model Zoo and Router Initialization.** In Mixture Bench, we experiment with `Which2` and `Which4` model settings. For router design, we mainly adopt a training-free linear layer router initialized from the prompt vector, as previous studies have demonstrated its effectiveness in the zero-shot MoE model [16]. For specific prompt settings, we refer to the Beyonder model series [4]. For the routing algorithm, we use Top-1 routing for `Which2` and *Block level mixture* and *Model-level mixture* for `Which4`, and Top-2 for `Which4` *FFN level mixture*.

Table 2: Model mixture methods and their abbreviations used in our study. Methods applicable for models with distinct architectures are highlighted in gray .

| Abbreviation | Mix. Level | Router | Router Input | Hybrid |
|---|---|---|---|---|
| F-L-T | FFN | Linear | Token | ✗ |
| Hybrid F-L-T | FFN | Linear | Token | ✓ |
| F-L-S | FFN | Linear | Sample | ✗ |
| F-M-S | FFN | MLP | Sample | ✗ |
| B-L-S | Block | Linear | Sample | ✗ |
| B-M-S | Block | MLP | Sample | ✗ |
| M-L-S | Model | Linear | Sample | ✗ |

**Post-mixture training.** For *MLP router* that are randomly initialized, we fine-tune the model by language modeling on the GPT4All dataset [5], only updating the router. We use the GPT4All [5] dataset for post-mixture router training, which is under Apache 2.0 License. For all the router training experiments, we apply the batch size of 128, a cosine learning rate scheduler, the learning rate of $5e - 5$, and the epochs of 1.

**Mixture Method Abbreviations.** To simplify the description, we use abbreviations to denote different mixture methods, as in Table 2.

### 4.5 Model Mixture Benchmark Results

In this section, we attempt to answer five main research questions about mixture variants: mixture level, router design, router input, and hybrid merging. We also explore the mixing of very different models that cannot be merged as the previous probe in our next `Model-GLUE` recipe that combines merging and blending for LLM scaling.

**Q1: At which level does the model mixture manifest its utmost effectiveness?**

**A: Model level mixture is consistently better.** Our comparative analysis of the {FFN, block, model} level mixture, all employing the linear router and the sample routing strategy as presented in Table 3, consistently demonstrates the superiority of the *Model level mixture* under `Which2` and `Which4` setting. This could be attributed to the design

Table 3: Comparison of different mixture levels. For each task in each model zoo, we highlight the performance best in each model zoo in **bold**.

| Model | ARC | WinoGrande | MMLU | GSM8K | MBPP | HumanEval | Average |
|---|---|---|---|---|---|---|---|
| | | | Which2 | | | | |
| Best Single Model | 54.27% | 71.51% | 47.24% | 21.30% | 18.00% | 13.06% | 37.68% |
| F-L-S | 52.82% | 70.80% | 50.04% | **23.12%** | 19.00% | 17.68% | 38.91% |
| B-L-S | 52.73% | 70.01% | 49.90% | 19.94% | 18.84% | 15.85% | 37.88% |
| M-L-S | **54.44%** | **72.38%** | **50.51%** | 22.21% | **20.00%** | **20.73%** | **40.04%** |
| | | | Which4 | | | | |
| Best Single Model | **55.03%** | 73.72% | **48.33%** | 24.26% | 17.80% | 13.41% | 38.70% |
| F-L-S | 53.75% | 73.88% | 47.97% | 34.87% | **21.80%** | **23.17%** | 42.57% |
| B-L-S | 52.65% | **74.66%** | 47.05% | 21.15% | 20.40% | 14.63% | 38.42% |
| M-L-S | 49.06% | 72.14% | 41.81% | **60.05%** | 17.60% | 15.24% | **42.65%** |

---

[4] https://huggingface.co/mlabonne/Beyonder-4x7B-v2

that *Model Level Mixture* route each sample to one expert model, thereby avoiding the conflicts between different expert models and maximizing the expertise of the most appropriate experts. Since the experts are not derived from the same pre-training process, directly merging their inconsistent representation spaces will affect the performance of the mixture model, with more expert parameters leading to worse results. This is especially evident for *Block-level Mixture*, as the routing is performed at each transformer layer and the representation is fed into different expert blocks in series, causing confusion when switching between different expert knowledge.

**Q2: Does more complex router design brings better results?**

**A: Not necessary, as the linear router outperforms the MLP router.** From Table 4, the performances of the *linear router* without additional training slightly surpass *MLP router* models, *i.e.*, F-L-T over F-M-T, B-L-S over B-M-S. Specifically, *linear router* models are better at math and coding datasets, validating prompt vector is effective in assorting samples from different domains, which is otherwise too implicit to learn via direct language modeling.

Table 4: Comparison between linear and MLP routers on `Which2` setting. We highlight better performance within each pair in **bold**.

| Model | ARC | WinoGrande | MMLU | GSM8K | MBPP | HumanEval | Average |
|---|---|---|---|---|---|---|---|
| F-L-T | 53.41% | 70.48% | **50.74%** | **23.28%** | **20.80%** | 16.46% | **39.20%** |
| F-M-T | **53.58%** | **72.06%** | 50.01% | 21.92% | 17.40% | **17.68%** | 38.78% |
| B-L-S | **52.73%** | 70.01% | **49.90%** | 19.94% | **18.84%** | **15.85%** | **37.88%** |
| B-M-S | 51.53% | **70.56%** | 49.41% | 19.94% | 16.60% | 14.02% | 37.01% |

**Q3: Does model mixture directly works on unmergeable models?**

**A: No.** We directly apply the setting of `Which2` *Model level mixture* to Llama-2-7b-chat and CrystalChat, an unmergeable model pair with different architectures and initialization. As shown in Table 5, the performance is slightly behind the best single model. This may be due to simple prompts and direct mixture, as it fails to coordinate the divergence between drastically different models. We evaluate more complex prompts for the same model pair and the mixture model outperforms, see Table 19 for more information.

Table 5: Comparison of the mixture of a unmergeable model pair (Llama-2-7b-chat and CrystalChat). We highlight the better performance in **bold**.

| Model | ARC | WinoGrande | MMLU | GSM8K | MBPP | HumanEval | Average |
|---|---|---|---|---|---|---|---|
| Best Single Model | **52.05%** | 69.46% | **50.77%** | 27.22% | **39.60%** | **35.98%** | **45.85%** |
| M-L-S | 50.68% | **69.77%** | 50.08% | **27.82%** | 33.80% | 30.48% | 43.77% |

**Q4: Which router input is better, token-level or sample-level?**

**A: Not quite different. Token input suits a mixture of the same domain models.** Table 6 shows the performance token-based and sample-based routing are pretty close. In particular, for `Which2` and `Which4` (`Chat`) where models are all trained for general chatting purposes, token routing outperforms, whereas sample routing is better for default `Which4` (`Domain`) with differently specialized models. This may result from divergence of model knowledge and representation spaces will cause conflicts in fine-grained token routing.

Table 6: Comparison of different router input designs. `Which4` includes one group with chatting models (`Chat`) and another with different domain models (`Domain`). We highlight the best performing mixture methods in **bold**.

| Model | ARC | WinoGrande | MMLU | GSM8K | MBPP | HumanEval | Average |
|---|---|---|---|---|---|---|---|
| | | | `Which2` | | | | |
| Best Single Model | **54.27%** | **71.51%** | 47.24% | 21.30% | 18.00% | 13.06% | 37.68% |
| F-L-T | 53.41% | 70.48% | **50.74%** | **23.28%** | **20.80%** | 16.46% | **39.20%** |
| F-L-S | 52.82% | 70.80% | 50.04% | 23.12% | 19.00% | **17.68%** | 38.91% |
| | | | `Which4` | | | | |
| Best Single Model | 55.03% | 73.72% | **48.33%** | 24.26% | 17.80% | 13.41% | 38.70% |
| Chat F-L-T | **55.63%** | 72.77% | 50.28% | 23.88% | 20.00% | **22.56%** | 40.85% |
| Chat F-L-S | 53.75% | 70.96% | 49.78% | 20.32% | **20.40%** | 20.12% | 39.22% |
| Domain F-L-T | **55.72%** | 74.11% | 48.32% | 30.17% | **22.00%** | 20.12% | 41.74% |
| Domain F-L-S | 53.75% | 73.88% | 47.97% | **34.87%** | 21.80% | **23.17%** | **42.57%** |

**Q5: Is it feasible for hybrid mixtures to provide enhancements?**

**A: Yes.** Our experiments on F-L-T with *v.s.* without the hybrid mixture, as detailed in Table 7, demonstrate that the hybrid mixture significantly improves performance on average and simultaneously reduces the memory overhead during inference. This improvement may be attributed to the

Table 7: Comparison between F-L-T methods with and without hybrid mixture technique. We highlight the best performing mixture methods in **bold**.

| Model | ARC | WinoGrande | MMLU | GSM8K | MBPP | HumanEval | Average |
|---|---|---|---|---|---|---|---|
| | | | `Which2` | | | | |
| Best Single Model | 54.27% | **71.51%** | 47.24% | 21.30% | 18.00% | 13.06% | 37.68% |
| F-L-T | 53.41% | 70.48% | **50.74%** | 23.28% | 20.80% | 16.46% | 39.20% |
| Hybrid F-L-T | **54.44%** | 71.19% | 50.45% | **23.96%** | **21.80%** | **18.29%** | **40.02%** |
| | | | `Which4` | | | | |
| Best Single Model | 55.03% | 73.72% | **48.33%** | 24.26% | 17.80% | 13.41% | 38.70% |
| F-L-T | **55.72%** | **74.11%** | 48.32% | 30.17% | 22.00% | 20.12% | 41.74% |
| Hybrid F-L-T | 54.86% | 73.80% | 48.23% | **37.53%** | **24.30%** | **23.17%** | **43.65%** |

higher sensitivity of the initial transformer blocks. Avoiding using MoE for these blocks can yield performance gains, as suggested by a few previous works as well [12, 41]. Surprisingly, our results show that the hybrid F-L-T model consistently outperforms the standard F-L-T on *math* and *code* tasks. Our further analysis indicates that this improvement might be because of the conversational nature of the content in GSM8K, MBPP, and HumanEval datasets, which appears to challenge the routing mechanisms within the initial transformer blocks, leading to ineffective expert specialization.

## 5 Superior Recipes to Aggregate LLM Knowledge

### 5.1 Model Merging *v.s.* Mixture

**Q1: For a mergeable model zoo, how should we choose between merging and mixture?** For limited computational resources and similar models, merging is always a simple and effective method. For the domain-specific models, mixture can bring greater improvements.

Detailed results are presented in Table 8. For `Which4 (Domain)`, due to the appropriately designed linear routers, model mixture can fully leverage various domain-specific models, thus slightly outperforming merging. For `Which4 (Chat)`, we adopt the optimal settings from `Which4 (Domain)` and only change the model zoo. Since individual models do not exhibit superior capabilities in a single domain, it is challenging to design suitable routers at a low cost. Therefore, mixture performed significantly worse compared to merging. Furthermore, although combining the homogeneous models in `Which4 (Chat)` brings some improvement, we can see that `Which4 (Domain)` overall outperforms `Which4 (Chat)`. Therefore, increasing the diversity among the models will make a greater contribution to the combined model.

Table 8: Comparison between the best merging approach *v.s.* the best mixture approach on `Which4 (Domain)` and `Which4 (Chat)`.

| Model | ARC | WinoGrande | MMLU | GSM8K | MBPP | HumanEval | Average |
|---|---|---|---|---|---|---|---|
| Best Single Model | 55.03% | 73.72% | 48.18% | 24.03% | 17.80% | 13.41% | 38.70% |
| Which4 (Domain) | | | | | | | |
| Merging | 54.01% | 73.64% | 47.39% | 43.75% | **22.40%** | **21.95%** | 43.86% |
| Mixture | **54.86%** | **74.11%** | **48.23%** | **49.81%** | 18.40% | 18.29% | **43.95%** |
| Which4 (Chat) | | | | | | | |
| Merging | **56.23%** | **73.72%** | **50.51%** | **25.85%** | **21.00%** | **21.95%** | **41.54%** |
| Mixture | 53.75% | 70.96% | 49.80% | 19.94% | 19.80% | 20.73% | 39.16% |

### 5.2 `Model-GLUE`: selective merging then model mixture for better LLM scaling

**Q2: How to combine models with greater differences in an extensive and varied model zoo?**

In `Which16`, a larger and more diverse model zoo , some models cannot be merged due to structural differences and models that would degrade in performance when merged with other models. Therefore, we first cluster the models based on cosine similarity. Within each mergeable family, we perform either merging or mixture. We initially employ heuristic strategies of merging and report the best results (*i.e.*, `Full Merging`) in Table 9. The Llama-2 family (*i.e.*, `Which12`) consists of up to 12 models, so directly combining them through the mixture is inefficient. Thus, we only consider models selected by merging and report the results of `F-L-T Mixture`. From Table 9, we can observe that `Full Merging` outperforms `F-L-T Mixture`.

Therefore, we selected `Full Merging` as the representative model for the Llama-2 family and combined it with other models that could not be merged by model mixture. On average, the `Model-GLUE` demonstrates a

Table 9: Comparison between the best single model, Full Merging, Full Mixture and our `Model-GLUE`.

| Model | ARC | WinoGrande | MMLU | GSM8K | MBPP | HumanEval | Average |
|---|---|---|---|---|---|---|---|
| Best Single Model | 46.76% | 64.33% | 46.33% | **62.40%** | 42.00% | 31.10% | 48.82% |
| Full Merging | **55.12%** | **73.64%** | 50.13% | 39.35% | 21.80% | 21.34% | 43.56% |
| F-L-T Mixture | 54.69% | 73.32% | 48.74% | 35.18% | 22.60% | 21.34% | 42.65% |
| Model-GLUE | 51.62% | 70.56% | **51.85%** | 53.53% | **47.20%** | **51.83%** | **54.43%** |

5.61% improvement over the `Best Single Model`. More details are presented in Appendix A.4.

## 6 Discussion with Other LLM Aggregation Techniques

Thus far, we mainly focus on two LLM aggregation techniques: model merging and mixture. In this section, we discuss other potential techniques that could help scaling existing LLMs.

**Model Stacking.** Research has demonstrated that stacking a model itself can accelerate training convergence as opposed to training a model of double the size from scratch [17, 18, 56, 60, 28]. This concept can be extended naturally to stack multiple models as one larger model. Our experimental results indicate that model stacking with lightweight fine-tuning can yield superior performance compared to various merging and mixture models. For instance, stacking 7B Llama-2-chat and Vicuna can achieve $\geq 55\%$ on the MMLU benchmark. When compared to model mixture, model

stacking offers less flexibility in terms of model choices. Although the resulting architecture is more standardized than MoE, increasing the model depth through stacking also results in higher latency than mixture models where subnetworks infer in parallel. Additionally, model stacking does not simplify the design space, such as determining whether, which, and how many layers should be dropped when stacking two heterogeneous models. We conducted a preliminary investigation employing model stacking techniques to address two primary research questions: (1) Can model stacking effectively combine the capabilities of two distinct models and surpass the performance of self-stacking a single model? (2) What is the impact of layer dropping on stacking performance?

Specifically, we examine the relationship between the number of dropped layers ($K$) and the resulting downstream task accuracy. To this end, we selected 7B Llama-2-Chat and Vicuna as the base models and fine-tuned the stacked models for 10 billion tokens. The obtained results are presented in Table 10. In the initial two rows, we report the performance of the two base models, revealing that Llama and Vicuna exhibit advantages on different datasets. In the subsequent two rows, we observe that stacking dissimilar models generally outperforms self-stacked models, and the weaknesses of one model can be compensated for by another stronger one. Moving forward, we explored the effects of varying the number of dropped layers. Our findings indicate that even when dumping half of each model ($K = 16$), the stacked 7B models can still significantly enhance performance across tasks.

Table 10: Comparison of different model stacking configurations.

| Model | ARC | WinoGrande | MMLU | Hellaswag | TruthfulQA |
|---|---|---|---|---|---|
| Llama-2-chat | 54.10% | 71.27% | 47.28% | 78.71% | 45.32% |
| Vicuna | 53.75% | 70.56% | 49.78% | 77.19% | 50.36% |
| Llama / Llama ($K = 8$) | 53.92% | 69.14% | 52.76% | 73.74% | 46.36% |
| Llama / Vicuna ($K = 8$) | 56.14% | 70.80% | 55.20% | 73.67% | 46.84% |
| Llama / Vicuna ($K = 12$) | 55.42% | 69.45% | 53.55% | 73.62% | 45.59% |
| Llama / Vicuna ($K = 16$) | 54.35% | 69.69% | 52.52% | 73.75% | 45.92% |
| Llama / Vicuna ($K = 20$) | 39.59% | 61.33% | 44.93% | 62.10% | 42.90% |
| Llama / Vicuna ($K = 24$) | 28.15% | 52.88% | 25.51% | 43.07% | 39.10% |

**Model Communication.** Model communication [61, 31, 33] is a framework that enables the development of LLM applications through the use of multiple conversable agents that collaborate to complete tasks. This approach allows developers to design complex LLM application workflows as multi-agent conversations, where agents with various roles and capabilities, driven by LLMs, tools, or human inputs, interact with each other. Unlike model merging, mixture, and stacking techniques, LLM communication is orthogonal to the primary focus of this paper because it does not modify the model weights; instead, it leverages the in-context learning and conversational capabilities of LLMs to coordinate agents. An empirical comparison with this class of methods is beyond the scope of this study and will be explored in future research.

## 7 Limitations

For LLM scaling studies, while empirical evidence suggests that increasing model size, data volume, and computational complexity leads to better performance, there is little theoretical clarity on the exact mechanisms behind these improvements. Second, although scaling laws suggest that performance continues to improve as models get larger, recent evidence indicates that scaling may lead to diminishing returns beyond a certain point. In addition, our work focuses on benchmarking results, while the reasons why model merging improves performance could be further enhanced by post hoc analysis, such as examining parameter distribution and similarity during model operations.

## 8 Conclusion

In this paper, we explore the scaling LLM based on a model zoo of pre-trained LLMs within the real world. We first benchmark state-of-the-art LLM merging, mixture, and model stacking. Based on previous findings, we then propose a novel LLM scaling framework, `Model-GLUE`. Specifically, we scale up the model zoo closely examine the existing model merging techniques, and conclude the selective merging techniques based on heuristics and learnable algorithms. Further, we investigate variants of Mixture-of-Experts for combining LLMs and suggest it can serve as an alternative to merging failure cases. Finally, we integrate selective merging strategies with model mixture techniques, presenting this as a comprehensive solution for scaling a diverse array of LLM collections. Future works will include model stacking and communication to our `Model-GLUE` framework.

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

# Appendix

## A  Implementation Details

### A.1  Detailed Algorithms of Heuristic Strategy of Model Merging

**Heuristic (Average).**    We present the implementation details in Algorithm 1. The algorithm takes a mergable model family as input and generate a merged model as output. For each candidate model in input model family, we compute the accuracy of the temporary merged model, generated by the union of this candidate model and the previously selected model, on the proxy dataset, and the candidate that brings no harm to the accuracy will be selected for the final merged model. Each weight of the merged model is generated by averaging the corresponding weights of all the selected models.

---

**Algorithm 1** Heuristic (Average)

---

**Input:** A mergable family $\{w_1, ..., w_n\}$ (sorted in decreasing order of $\text{Acc}(w_i)$).
**Output:** *merged_model*
 1: *models_to_merge* $\leftarrow \{w_1\}$
 2: *merged_model* $\leftarrow w_1$
 3: **for** $i = 2$ to $n$ **do**
 4:     **if** ProxyAcc(AvgMerge(*models_to_merge* $\cup \{w_i\}$)) $\geq$ ProxyAcc(*merged_model*)) **then**
 5:         *models_to_merge* $\leftarrow$ *models_to_merge* $\cup \{w_i\}$
 6:         *merged_model* = AvgMerge(*models_to_merge*)
 7:     **end if**
 8: **end for**
 9: **return** *merged_model*

---

**Heuristic (Coefficient).**    We present the implementation details in Algorithm 2. Heuristic (Coefficient) builds upon Heuristic (Average) by combining the previously merged model with a new candidate using different coefficients in each round. To reduce the search space, we set the range of coefficient as 0.1, 0.2...0.9.

---

**Algorithm 2** Heuristic (Coefficient)

---

**Input:** A mergable family $\{w_1, ..., w_n\}$ (sorted in decreasing order of $\text{Acc}(w_i)$), a list of coefficients $\{0.1, 0.2..., 0.9\}$ to be searched when merging.
**Output:** merged_model
 1: *coefficients* $\leftarrow \{0.1, 0.2..., 0.9\}$
 2: *merged_model* $\leftarrow w_1$
 3: **for** $i = 2$ to $n$ **do**
 4:     *best_acc*, *best_c* $\leftarrow$ ProxyAcc(*merged_model*), 1.0
 5:     **for** $c$ in *coefficients* **do**
 6:         **if** ProxyAcc(Merge($c$, *merged_model*, $w_i$)) $\geq$ *best_acc* **then**
 7:             *best_acc*, *best_c* $\leftarrow$ ProxyAcc(Merge($c$, *merged_model*, $w_i$)) , $c$
 8:         **end if**
 9:     **end for**
10:     *merged_model* $\leftarrow$ Merge(*best_c*, *merged_model*, $w_i$)
11: **end for**
12: **return** *merged_model*

---

**Heuristic (Similarity).**    We present the implementation details in Algorithm 3. We use the average similarity of all weights as the criterion for selecting models in each round. This algorithm selects the candidate model with the highest or lowest similarity and conducts a coneflicient search to combine it with the previously merged model.

---
**Algorithm 3** Heuristic (Similarity)
---
**Input:** A mergable family $\{w_1, ..., w_n\}$ (sorted in decreasing order of $\text{Acc}(w_i)$), a list of coefficients $\{0.1, 0.2..., 0.9\}$ to be searched when merging.
**Output:** merged_model
 1: $merged\_model \leftarrow w_1$
 2: $remaining\_models \leftarrow \{w_2, ..., w_n\}$
 3: **for** $i = 2$ to $n$ **do**
 4:     $best\_acc, best\_c \leftarrow \text{ProxyAcc}(merged\_model), 1.0$
 5:     $candidate\_model \leftarrow \text{GetModelBySimilarity}(merged\_model, remaining\_models)$
 6:     **for** $c$ in *coefficients* **do**
 7:         **if** $\text{ProxyAcc}(\text{Merge}(c, merged\_model, candidate\_model)) \geq best\_acc$ **then**
 8:             $best\_acc, best\_c \leftarrow \text{ProxyAcc}(\text{Merge}(c, merged\_model, candidate\_model)) , c$
 9:         **end if**
10:     **end for**
11:     $merged\_model \leftarrow \text{Merge}(best\_c, merged\_model, candidate\_model)$
12:     $remaining\_models \leftarrow remaining\_models \setminus \{candidate\_model\}$
13: **end for**
14: **return** $merged\_model$
---

## A.2 Detailed about Evolutionary Strategy of Model Merging

For the experiments of **Q2** - (*i*) in Section 4.3, we constrain all parameter values to be within the range of $[0, 1]$. TIES and DARE require to optimize $2 * k$ parameters, while other methods require to optimize $k$ parameters, where $k$ represents the number of models included in the model zoo.

For the experiments of **Q2** - (*ii*) in Section 4.3, we choose the Linear method for experimentation, and we constrain all parameter values to be within the range of $[0, 1]$. For finer-grained merging, we group adjacent $n$ decoder layers together, where they share the same coefficient. For the remaining parameters, we make them share the same coefficient. Hence, the number of parameters that need to be fine-tuned is given by: $k * (\frac{num\_hidden\_layers}{n} + 1)$, where $k$ represents the number of models and $n$ represents the size of groups. For the case of $n = 32$, we utilized the previous results, thus the number of parameters to be optimized is $k$.

For the experiments of **Q2** - (*iii*) in Section 4.3, we control the variation of coefficients obtained through heuristic strategy to not exceed $0.1$, and when it is negative, we set it to $0$. We also only evaluate the Linear method.

## A.3 Detailed Algorithms of Model Mixture

**Model Level Mixture.** We present the implementation details in Algorithm 4. The mixed model consists of a router, which determines the expert to execute inference, and all the input models as experts. All the weights of input model's components, including embedding layers (embd_layer), decoder layers (layers) and language model head (lm_head), will be integrated into the mixed model.

---
**Algorithm 4** Model Level Mixture
---
**Input:** A model family $\{w_1, ..., w_n\}$
**Output:** *mixed_model*
 1: $mixed\_model.router \leftarrow \text{GenerateRouter}(\{w_1, ..., w_n\})$
 2: **for** $i = 1$ to $n$ **do**
 3:     $mixed\_model.expert_i \leftarrow w_i$
 4: **end for**
 5: **return** $mixed\_model$
---

**Block Level Mixture.** We present the implementation details in Algorithm 5. Different from model-level mixture, block-level mixture utilizes the embd_layer and lm_head of an additional model within a model family to handle input and output. Meanwhile, the transformer blocks of other models within the model family act as experts, connected by a router.

**FFN Level Mixture.** We present the implementation details in Algorithm 6. FFN level mixture is similar to block level with only difference on inner-block component sharing. Each layer of the

---

**Algorithm 5** Block Level Mixture

---

**Input:** A model family $\{w_1, ..., w_n\}$ with identical layer amount, one of the family as $base\_model$
**Output:** $mixed\_model$
 1: $mixed\_model.embd\_layer \leftarrow base\_model.embd\_layer$
 2: $mixed\_model.lm\_head \leftarrow base\_model.lm\_head$
 3: **for** $i = 0$ to Len($base\_model.layers$) **do**
 4:     $mixed\_model.layer_i.router \leftarrow$ GenerateRouter($\{w_1, ..., w_n\}$)
 5:     **for** $j = 1$ to $n$ **do**
 6:         $mixed\_model.layer_i.expert_j \leftarrow w_j.layer_i$
 7:     **end for**
 8: **end for**
 9: **return** $mixed\_model$

---

mixed model will take the attention weights of the base model and build an MoE structure based on the FFNs in corresponding layers of all the input models.

---

**Algorithm 6** FFN Level Mixture

---

**Input:** A model family $\{w_1, ..., w_n\}$ with identical layer amount, one of the family as $base\_model$.
**Output:** $mixed\_model$,
 1: $mixed\_model.embd\_layer \leftarrow base\_model.embd\_layer$
 2: $mixed\_model.lm\_head \leftarrow base\_model.lm\_head$
 3: **for** $i = 0$ to Len($base\_model.layers$) **do**
 4:     $mixed\_model.layer_i.router \leftarrow$ GenerateRouter($\{w_1, ..., w_n\}$)
 5:     $mixed\_model.layer_i.attention \leftarrow base\_model.layer_i.attention$
 6:     $mixed\_model.layer_i.norm \leftarrow base\_model.layer_i.norm$
 7:     **for** $j = 1$ to $n$ **do**
 8:         $mixed\_model.layer_i.expert_j \leftarrow w_j.layer_i.FFN$
 9:     **end for**
10: **end for**
11: **return** $mixed\_model$

---

**Hybrid Mixture**    We present the implementation details in Algorithm 7. The hybrid mixture combines both merging and mixture methods. Specifically, the first $k$ layers of the mixed model are obtained by merging multiple models, while the rest of the layers use an FFN-level mixture architecture.

---

**Algorithm 7** Hybrid Mixture

---

**Input:** A model family $\{w_1, ..., w_n\}$ with identical layer amount, one of the family as $base\_model$,
    $k$ layers for merging and the rest layers for mixture.
**Output:** $mixed\_model$
 1: $mixed\_model.embd\_layer \leftarrow base\_model.embd\_layer$
 2: $mixed\_model.lm\_head \leftarrow base\_model.lm\_head$
 3: **for** $i = 0$ to $k$ **do**
 4:     $mixed\_model.layer_i \leftarrow$ Merge($\{w_1, ..., w_n\}$, i)
 5: **end for**
 6: **for** $i = k + 1$ to Len($base\_model.layers$) **do**
 7:     $mixed\_model.layer_i.router \leftarrow$ GenerateRouter($\{w_1, ..., w_n\}$)
 8:     $mixed\_model.layer_i.attention \leftarrow base\_model.layer_i.attention$
 9:     $mixed\_model.layer_i.norm \leftarrow base\_model.layer_i.norm$
10:     **for** $j = 1$ to $n$ **do**
11:         $mixed\_model.layer_i.expert_j.FFN \leftarrow w_j.layer_i.FFN$
12:     **end for**
13: **end for**
14: **return** $mixed\_model$

---

Table 11: Performance of merged models with different similarity. Sim. stands for cosine similarity.

| Parent Model 1 | Parent Model 2 | ARC | MMLU | WinoGrande | GSM8K | HumanEval | MBPP | Avg. | Sim. |
|---|---|---|---|---|---|---|---|---|---|
| Llama-2-7b-hf | deepseek-coder-6.7b-base | 27.73% | 24.38% | 49.64% | 0.00% | 0.00% | 0.00% | 16.96% | 0% |
| Llama-2-7b-hf | CodeLlama-7b-hf | 41.04% | 31.68% | 66.85% | 5.76% | 10.98% | 21.40% | 29.62% | 52.55% |
| CodeLlama-7b-Python-hf | CodeLlama-7b-hf | 40.61% | 37.17% | 65.35% | 6.67% | 21.95% | 25.60% | 32.89% | 60.34% |
| MetaMath-Llemma-7B | CodeLlama-7b-hf | 46.16% | 42.86% | 64.64% | 27.07% | 34.76% | 37.40% | 42.15% | 88.70% |
| CodeLlama-7b-Instruct-hf | CodeLlama-7b-hf | 43.86% | 41.39% | 68.59% | 16.07% | 33.54% | 40.80% | 40.71% | 99.94% |

## A.4 Details of Model-Glue

The models selected by the heuristic strategy include: migtissera/Synthia-7B-v1.2, neuralmagic/Llama-2-7b-evolcodealpaca, teknium/OpenHermes-7B, meta-llama/Llama-2-7b-chat-hf, meta-math/MetaMath-7B-V1.0, lmsys/vicuna-7b-v1.5. Since merging ise-uiuc/Magicoder-S-CL-7B and codellama/CodeLlama-7b-Instruct-hf does not lead to improvement in the Codellama's mergeable family, we select ise-uiuc/Magicoder-S-CL-7B as the representative model.

The final models used for Model-level Mixture are: LLM360/CrystalChat, ise-uiuc/Magicoder-S-CL-7B, meta-math/MetaMath-Llemma-7B and the representative model of the Llama-2 family obtained through the Heuristic (Coefficient). Please refer to our repository for specific configurations.

## A.5 Details of clustering in selective merging pipeline

**Motivation for using cosine similarity as a model selection criterion**   Previous merging study [64] finds that merging performance is consistent with parameter similarity. We inherit it by using cosine similarity as a representative method to measure whether a model can be merged. From our preliminary result, cosine similarity works effectively. Empirically, when the cosine similarity between models exceeds 0.95, merging them can yield positive benefits. In Table 14, we present examples of successful and unsuccessful merging. For example, the cosine similarity between the weights of Llama-2-chat and Vicuna is 0.9982, resulting in the merged model significantly outperforming its parent models. On the other hand, the cosine similarity between the weights of Llama-2-chat and CodeLlama is 0.5351, indicating that the merged model is inferior to CodeLlama. Moreover, using cosine similarity to measure the merging benefit is simple and efficient. For these reasons, we stick with cosine similarity for selective merging pipelines.

**Criteria for Determining the Number of Clusters.**   We cluster models with cosine similarity greater than 0.95 into a mergeable family, ensuring that within this mergeable family, the pairwise similarities between models are greater than 0.95. The number of clusters is automatically determined during the process, after which we execute our merge strategy within each cluster. For `Which16` model zoo in our paper, we clustered 16 models and finally obtained five mergeable families: ❶ 12 models fine-tuned based on llama-2, ❷ ise-uiuc/Magicoder-S-CL-7B, ❸ codellama/CodeLlama-7b-Instruct-hf, ❹ meta-math/MetaMath-Llemma-7B, ❺ LLM360/CrystalChat. Since the remaining clusters contain only one model each, we only report the results of different merging strategies performed within Family ❶.

**Impact of clustering threshold**   We computed the cosine similarity between 12 LLMs all fine-tuned from Llama-2. These models are considered to be well mergeable, having the same architecture and initialization. Since their similarities range from 0.9680 to 0.9999, 0.95 could be a lower bound for model clustering. To show the impact of different clustering thresholds, we have examined the performance of merged models with drastically different similarity: Llama-2,deepseek-coder, CodeLlama, and MetaMath-Llema. We use linear interpolation to merge two models and present the benchmarking results in Table 11. The performance of the individual models is shown in Table 12. If the merged model outperforms its parent models on average accuracy, we consider it a successful merge. From Table 11, we see that successful merging only occurs between Codellama and Codellama-instruct whose weights reach 0.99 similarity and have the same initialization. To include more mergeable models, we finally choose 0.95 as the threshold for clustering.

## A.6 Energy Consumption

Existing literature is mainly concerned with carbon emissions during LLM pre-training [14, 52]. However, the training costs associated with the approaches evaluated in our benchmark are minimal. Specifically, the only training expenditure in our study pertains to the B-M-S router training, as described in Section 4.4. This process requires about 80 GPU hours, resulting in 13.55kg $CO_2$

Table 12: Performance of parent models.

| Model | ARC | MMLU | WinoGrande | GSM8K | HumanEval | MBPP | Avg. |
|---|---|---|---|---|---|---|---|
| Llama-2-7b-hf | 53.92% | 45.83% | 74.11% | 13.72% | 10.98% | 18.00% | 36.09% |
| deepseek-coder-6.7b-base | 36.86% | 36.36% | 57.30% | 19.03% | 45.12% | 54.80% | 41.58% |
| CodeLlama-7b-hf | 41.89% | 39.05% | 65.98% | 11.83% | 32.32% | 37.20% | 38.05% |
| CodeLlama-7b-Python-hf | 40.70% | 35.62% | 64.56% | 13.12% | 38.41% | 41.20% | 38.94% |
| MetaMath-Llemma-7B | 46.67% | 46.29% | 64.33% | 62.24% | 32.32% | 42.00% | 48.97% |
| CodeLlama-7b-Instruct-hf | 43.00% | 41.69% | 65.90% | 18.12% | 33.70% | 40.00% | 40.40% |

Table 13: Comparison between the best single model, Merging, Full Mixture and our Model-GLUE with Mistral model zoo. We highlight the better performance in **bold**.

| Model | ARC | WinoGrande | MMLU | GSM8K | MBPP | HumanEval | Average |
|---|---|---|---|---|---|---|---|
| Best Single Model | **67.24**% | 79.01% | 61.77% | 63.15% | 35.98% | 39.00% | 57.69% |
| DARE | 64.33% | 78.37% | 63.27% | 63.31% | 39.02% | **44.60**% | 58.82% |
| TIES | 63.74% | 77.90% | 60.90% | 49.13% | 34.76% | 39.40% | 54.30% |
| F-L-S | 64.85% | **79.72**% | 63.42% | 64.82% | 42.00% | 42.07% | 59.48% |
| Model-GLUE | 65.02% | 78.85% | **64.39**% | 65.50% | **44.60**% | 42.68% | **60.18**% |

emissions based on a 400W power consumption. In contrast, LLaMA-2-7B pre-training results in 31.22t CO2, which is over 2000 times more than ours.

# B  Additional Results

## B.1  Experiment on Mistral model family

We choose the Llama2-based model family for the main experiments because there are more diverse variances built on different datasets and training recipes. There are many domain-specific models based on Llama-2, such as those for code, mathematics, healthcare, finance, law, and mental health. Importantly, a series of models have undergone continuous pre-training based on Llama-2, and a considerable portion of models trained from scratch have drawn inspiration from the architecture of Llama-2. While these models share the same architecture as Llama-2, their weights exhibit significant differences. Thus, we can thoroughly examine the effect of merging, mixture and Model-GLUE on different settings. To further evaluate our proposal on the Mistral model family, we have established a Mistral-based `Which8` model zoo and replicated the experiments outlined in Section 5.2. From the result in Table 13 it can be seen that `Mode-GLUE` consistently outperform.

## B.2  Model Merging

We present the specific results of Figure 4 in Table 15 and Table 16 and other results of Section 4.3 in Table 14, Table 17 and Table 18.

## B.3  Model Mixture

For Model Level Mixture, we use more fine-grained prompts to construct the router, and report the results in Table 19.

Table 14: Failure case of existing merging approaches when expanding the model zoo.

| Merging Method | ARC | WinoGrande | MMLU | GSM8K | MBPP | HumanEval | Average |
|---|---|---|---|---|---|---|---|
| Single Model | | | | | | | |
| Llama-2-chat | 54.10% | 71.27% | 47.28% | 23.05% | 17.00% | 13.41% | 37.68% |
| Vicuna | 53.75% | 70.56% | 49.78% | 19.11% | 6.00% | 19.51% | 36.45% |
| CodeLlama | 43.52% | 65.11% | 41.83% | 17.06% | 40.00% | 33.70% | 40.20% |
| Merge Llama-2-chat and Vicuna | | | | | | | |
| Linear | 54.27% | 72.30% | **50.72%** | 24.49% | 20.80% | **20.12%** | 40.45% |
| Model Stock | 54.61% | **74.43%** | 47.44% | 16.07% | **22.40%** | 14.02% | 38.16% |
| SLERP | **55.29%** | 72.45% | 50.51% | 24.87% | 21.80% | **20.12%** | **40.84%** |
| Task Arithmetic | 54.27% | 71.67% | 49.95% | 26.31% | 21.40% | 17.07% | 40.11% |
| DARE | 54.35% | 72.14% | 50.38% | **26.61%** | 21.00% | 17.68% | 40.36% |
| TIES | 52.65% | 69.93% | 49.84% | 24.34% | 17.60% | 19.51% | 38.98% |
| Merge Llama-2-chat and CodeLlama | | | | | | | |
| Linear | 45.05% | 67.09% | 39.03% | 16.76% | **36.60%** | **23.17%** | 37.95% |
| Model Stock | 50.34% | 71.27% | 41.06% | 10.01% | 15.40% | 7.93% | 32.67% |
| SLERP | **52.05%** | **71.43%** | **46.41%** | 18.95% | 20.80% | 18.90% | **38.09%** |
| Task Arithmetic | 44.97% | 68.03% | 38.83% | 7.05% | 10.60% | 12.20% | 30.28% |
| DARE | 38.91% | 65.98% | 31.90% | 3.34% | 15.00% | 9.76% | 27.48% |
| TIES | 21.67% | 49.88% | 25.25% | 0.00% | 0.00% | 0.00% | 16.13% |

Table 15: Comparison between different Heuristic Strategies.

| Heuristic Strategy | ARC | WinoGrande | MMLU | GSM8K | MBPP | HumanEval | Average |
|---|---|---|---|---|---|---|---|
| Best Single Model | 55.03% | 73.72% | 48.18% | 24.03% | 17.80% | 13.41% | 38.70% |
| Which12 | | | | | | | |
| Average | 54.86% | 73.48% | 49.42% | 32.98% | **23.60%** | **21.34%** | 42.61% |
| Coefficient | 55.12% | **73.64%** | 50.13% | 39.35% | 21.80% | **21.34%** | **43.56%** |
| Similarity↑ | **56.48%** | 73.32% | **52.56%** | 37.91% | 17.80% | 20.12% | 43.03% |
| Similarity↓ | 55.80% | 71.74% | 52.39% | **47.99%** | 16.40% | 15.85% | 43.36% |
| Which8 | | | | | | | |
| Average | **55.38%** | **74.11%** | 48.65% | 34.42% | **25.20%** | **23.17%** | 43.49% |
| Coefficient | 55.12% | 73.64% | **50.13%** | 39.35% | 21.80% | 21.34% | **43.56%** |
| Similarity↑ | 54.95% | 73.64% | 49.00% | 43.75% | 19.80% | 11.59% | 42.12% |
| Similarity↓ | 54.78% | 72.30% | 49.06% | **47.23%** | 21.20% | 15.85% | 43.40% |
| Which4 | | | | | | | |
| Average | 54.86% | 73.16% | 47.91% | 37.00% | **24.00%** | **21.34%** | 43.05% |
| Coefficient | **55.12%** | **74.03%** | **48.18%** | 41.93% | 19.40% | 14.63% | 42.22% |
| Similarity↑ | 54.52% | 73.24% | 47.81% | 41.77% | 21.20% | 20.73% | **43.21%** |
| Similarity↓ | 53.92% | 73.56% | 47.81% | **48.45%** | 18.20% | 10.98% | 42.15% |

Table 16: Comparison between different merging methods.

| Merging Method | ARC | WinoGrande | MMLU | GSM8K | MBPP | HumanEval | Average |
|---|---|---|---|---|---|---|---|
| Which12 | | | | | | | |
| Linear | **56.48%** | **73.56%** | 51.79% | 36.01% | **23.60%** | **20.12%** | **43.59%** |
| Task Arithmetic | 51.54% | 69.14% | 51.07% | **54.66%** | 1.80% | 13.41% | 40.27% |
| DARE | 51.19% | 70.09% | 51.03% | 53.53% | 6.80% | 12.80% | 40.91% |
| TIES | 53.75% | 70.64% | **52.77%** | 49.36% | 17.20% | 2.44% | 41.03% |
| Which8 | | | | | | | |
| Linear | **55.12%** | **73.64%** | **49.59%** | 40.64% | **22.40%** | 18.90% | 43.38% |
| Task Arithmetic | 52.65% | 70.64% | 48.11% | 51.18% | 19.80% | **21.95%** | **44.05%** |
| DARE | 52.56% | 71.19% | 49.00% | **53.37%** | 14.80% | 20.12% | 43.51% |
| TIES | 50.26% | 71.27% | 48.58% | 47.69% | 18.40% | 0.61% | 39.47% |
| Which4 | | | | | | | |
| Linear | **53.50%** | **73.01%** | **47.32%** | 45.79% | **20.20%** | 15.85% | 42.61% |
| Task Arithmetic | 52.73% | 72.30% | 46.81% | **51.86%** | 18.20% | 18.29% | **43.36%** |
| DARE | 51.45% | 71.67% | 45.61% | 51.55% | 16.60% | **20.12%** | 42.83% |
| TIES | 50.51% | 71.98% | 46.62% | 49.43% | 16.40% | 1.22% | 39.36% |

Table 17: The impact of different group sizes on Evolutionary Strategy.

| Group Size | ARC | WinoGrande | MMLU | GSM8K | MBPP | HumanEval | Average |
|---|---|---|---|---|---|---|---|
| | | | Which12 | | | | |
| 1 | 56.14% | 73.32% | 51.50% | 32.45% | **24.00%** | **20.12%** | 42.92% |
| 4 | 56.31% | 73.72% | 52.04% | 33.43% | **24.00%** | 18.90% | 43.07% |
| 8 | **56.83%** | **74.43%** | **53.01%** | **38.13%** | 21.20% | 19.51% | **43.85%** |
| 32 | 56.48% | 73.56% | 51.79% | 36.01% | 23.60% | **20.12%** | 43.59% |
| | | | Which8 | | | | |
| 1 | 55.12% | 74.11% | 49.96% | 31.69% | **25.20%** | 19.51% | 42.60% |
| 4 | **56.06%** | **74.66%** | **50.04%** | 33.59% | 24.20% | **21.95%** | 43.42% |
| 8 | 55.29% | 73.88% | 49.20% | 40.56% | 24.60% | 18.90% | **43.74%** |
| 32 | 55.12% | 73.64% | 49.59% | **40.64%** | 22.40% | 18.90% | 43.38% |
| | | | Which4 | | | | |
| 1 | **54.61%** | 73.32% | **47.63%** | 41.62% | 23.60% | 15.85% | 42.77% |
| 4 | 52.90% | 73.32% | 46.99% | 43.06% | **24.00%** | 20.73% | 43.50% |
| 8 | 54.01% | **73.64%** | 47.39% | 43.75% | 22.40% | **21.95%** | **43.86%** |
| 32 | 53.50% | 73.01% | 47.32% | **45.79%** | 20.20% | 15.85% | 42.61% |

Table 18: More efficient merging strategy.

| Strategy | ARC | WinoGrande | MMLU | GSM8K | MBPP | HumanEval | Average | Round |
|---|---|---|---|---|---|---|---|---|
| | | | Which12 | | | | | |
| Evo (Vanilla) | 56.48% | 73.56% | 51.79% | 36.01% | 23.60% | 20.12% | 43.59% | 200 |
| Evo (Heuristic) | 55.29% | 72.85% | 49.96% | 40.56% | 22.80% | 18.29% | 43.29% | 127 |
| | | | Which8 | | | | | |
| Evo (Vanilla) | 55.12% | 73.64% | 49.59% | 40.64% | 22.40% | 18.90% | 43.38% | 200 |
| Evo (Heuristic) | 54.69% | 72.93% | 49.68% | 45.19% | 21.00% | 19.51% | 43.83% | 71 |
| | | | Which4 | | | | | |
| Evo (Vanilla) | 53.50% | 73.01% | 47.32% | 45.79% | 20.20% | 15.85% | 42.61% | 200 |
| Evo (Heuristic) | 54.52% | 73.56% | 47.74% | 40.71% | 23.20% | 21.95% | 43.61% | 69 |

Table 19: Better prompt vector for the mixture of Llama-2-7b-chat and CrystalChat. We highlight the better performance in **bold**.

| Model | ARC | WinoGrande | MMLU | GSM8K | MBPP | HumanEval | Average |
|---|---|---|---|---|---|---|---|
| Best Single Model | 52.05% | 69.46% | 50.77% | 27.22% | **39.60%** | **35.98%** | 45.85% |
| M-L-S | **51.88%** | **70.88%** | **52.44%** | **32.52%** | 39.40% | 31.10% | **46.37%** |

