# OpenReview forum: "$\texttt{Model-GLUE}$: Democratized LLM Scaling for A Large Model Zoo in the Wild"
_NeurIPS.cc/2024/Datasets_and_Benchmarks_Track — NeurIPS 2024 Track Datasets and Benchmarks Poster_

### Official Review · Reviewer_9iRC · 2024-06-27
**Interesting benchmark study of how to combine LLMs**

**Rating:** 6
**Confidence:** 2

**Review:**

Due to the expense of training LLMs it is becoming increasingly popular to build new models by combining existing models with potentially complementary capabilities. This is a theoretically un-understood area of research that currently relies on various heuristics. The key question addressed by this work is then how to best navigate this growing space of heuristics. The paper considers a fairly large collection of models and empirically investigate the performance of different strategies on several tasks.

**Strengths:**

Strengths:
* The paper performs a quite thorough study on many combinations of strategies.
* The results are summarized in an easily accessible Q&A form, which I find very valuable.
* The paper provides a current-best-practice strategy in the form of the proposed Model-GLUE.

**Additional Feedback:**

I do not understand what is meant by "Democratized" in the paper title. What is democratic about the paper? (also note that on OpenReview the paper title contains LaTeX commands)

**Clarity:**

The paper is generally well-written. I have a series of minor corrections below, but nothing that needs addressing in the rebuttal.

In several places in the paper, we have text like "[58] first propose...", which is difficult to read (a number did not propose anything). I recommend rephrasing or using the \citet command to get an author name.

Fig 1 caption: typo "Model-GLEU"

Line 174: "mixture-of-mixture", I am unsure if this is a typo (i.e. should it have been "mixture-of-experts") or if this is deliberate. At least the term is never used elsewhere in the paper.

Line 193: "rout" --> "route" ?

Line 226: "trails" --> "trials" ?

I find it confusing when the paper talks about "clustering" of models by architecture (e.g. line 219). In my mind, "clustering" refers to something like k-means; in this case, I assume the authors refer to grouping by architecture, which sounds more like something implemented with a collection of if-statements.

**Correctness:**

I have no concerns regarding correctness, but I note that I do not work with LLMs so I cannot provide expert evaluation.

**Documentation:**

A GitHub link is provided in the paper, but at the time of reading the paper, this repository is empty. This is highly problematic.

**Ethics:**

I have no ethical concerns here.

**Limitations:**

I miss a discussion of energy usage of the resulting models. Ideally, this should have been part of the benchmark (which exclusively focuses on how well the models work), but, at the least, the paper should have discussed energy usage. Given the environmental impact of the vast deployment of LLMs, I think this is essential.

**Opportunities For Improvement:**

* One can always request a large set of models, heuristics, and datasets. I am not an expert in this domain, so I cannot judge if such requests are reasonable considering the combinatorial growth in required compute resources.
* I wished the paper had done more to address the energy consumption of the resulting models. Different combination heuristics will lead to models with different energy levels, which is both an environmental and financial concern. I think current heuristics should also be ranked accordingly.

**Relation To Prior Work:**

The paper discusses related work, but I am not familiar with ongoing work in the field, so I cannot determine if this is sufficient.

**Summary And Contributions:**

The paper performs an empirical evaluation of how to best combine multiple large language models (LLMs) to improve performance across tasks. This includes a quite thorough comparison of model merging, stacking, and mixture of experts. The results are summarized in an easily accessible Q&A form.

---

> ### Author Rebuttal · Authors · 2024-08-17
>
> We are very glad and appreciate that reviewer 9iRC had a positive first impression. To address reviewer 9iRC’s questions, we provide point-by-point responses to your concerns as follows.
>
> **[QUESTION1 Requesting a large set of models, heuristics, and datasets]**.
>
> - We have included a "large set of models, heuristics, and datasets" in our paper, which is summarized here.
> We conducted an extensive benchmarking analysis of LLM merging and model mixture strategies. Our study of model mixtures spans four dimensions: mixture level, router design, router input, and hybrid mixture strategies.
> - Our benchmark includes 16 open-source LLMs organized into five groups and six well-known benchmark datasets across three categories, ensuring a thorough evaluation. In addition, our LLM merging benchmark includes three heuristics and one evolutionary strategy.
> - Based on our benchmark evaluations, we have formulated an optimal strategy for selecting and combining a diverse collection of models that differ in architecture and initialization into an effective model ensemble.
> To address your concern, we have performed additional experiments with the Mistral model family, setting up a Mistral-based Which8 model zoo and repeating the experiments described in Section 5.2. The result below shows that Mode-GLUE consistently outperforms.
>
> Table S1 Comparison between the best single model, Merging, Full Mixture and our Model-GLUE with Mistral model zoo
> |                             | ARC       | WinoGrande | MMLU      | GSM8K | MBPP      | HumanEval | Avg       |
> | --------------------------- | --------- | ---------- | --------- | ----- | --------- | --------- | --------- |
> | Best single model           | **67.24** | 79.01      | 61.77     | 63.15 | 35.98     | 39.00     | 57.69     |
> | DARE Merging [1]              | 64.33     | 78.37      | 63.27     | 63.31 | 39.02     | **44.60** | 58.82     |
> | TIES Merging [2]               | 63.74     | 77.90      | 60.90     | 49.13 | 34.76     | 39.40     | 54.30     |
> | FFN-Linear-Sentence Mixture | 64.85     | **79.72**  | 63.42     | 64.82 | 42.00     | 42.07     | 59.48     |
> | Model-GLUE                  | 65.02     | 78.85      | **64.39** | 65.50 | **44.60** | 42.68     | **60.18** |
> - We will include more details in the final paper version as suggested by the reviewer.
>
> **[QUESTION2 Energy Consumption]**.
> Thank you for your suggestion. To the best of our knowledge, the existing literature is mainly concerned with carbon emissions during LLM pre-training [1, 2]. However, the training costs associated with the approaches evaluated in our benchmark are minimal. Specifically, the only training expenditure in our study pertains to the B-M-S model’s MLP router training, as described in Section 4.4. This process requires about 80 GPU hours, resulting in 13.55 kgCO2 emissions based on a 400W power consumption. In contrast, LLaMA-2-7B's pre-training results in 31.22 tCO2, which is over 2000 times more than ours. We will include these results and additional clarifications in the revised version of our paper.
>
> **[QUESTION3 Code issue]**.
> Thank you for the detailed review, your question will greatly improve our code management. Our complete codes are maintained in [model merging](https://github.com/Model-GLUE/model_merge_llm), [model mixture](https://github.com/Model-GLUE/model_mixture_llm), and [model stacking](https://github.com/Model-GLUE/model_stack_llm), all of which were committed prior to paper submission. We accidentally put the wrong link in the abstract. We will include it in a repository.
>
> **[QUESTION4 Writing Details]**.
> Thanks for your kind reminder, we will use the "\citet" command to get an author name and include it in our revised paper. As for the typos in our current manuscript, we will fix them and include them in our revised paper.
>
> **[QUESTION5 Confusion about "clustering" models by architecture]**.
> The "clustering" algorithm in our paper is used to group models. Specifically, we first divide those models with different architectures into different groups. Then, for those models with the same architecture, we cluster them based on cosine similarity with a threshold of 0.95. As a result, models with the same architecture may end up in different clusters due to low cosine similarity of their weights. For example, "lmsys/vicuna-7b-v1.5" and "meta-math/MetaMath-Llemma-7B" share the same architecture, but the former belongs to the Which-12 model family while the latter does not.
>
> **[QUESTION6 Meaning of "democratized" in the paper title]**.
> The term "democratized" in this context emphasizes the goal of making LLM scaling more accessible. This involves the adoption of several widely used techniques that we have integrated into 'ModelGLUE' - a novel paradigm for LLM scaling that facilitates the seamless integration of a wide variety of models in practical applications.
>
> [1] Measuring the Carbon Intensity of AI in Cloud Instances, ACM FAccT 2022
>
> [2] Llama 2: Open Foundation and Fine-Tuned Chat Models, arXiv:2307.09288

---

### Official Review · Reviewer_5WA2 · 2024-07-26
**Analysis of Model Merging and Mixture-of-Experts Methods in Llama-2-Based Model Zoo**

**Rating:** 6
**Confidence:** 4
**Correctness:** Please kindly see above.
**Clarity:** I think the overall message conveyed …

**Review:**

- The paper and benchmarks are solely based on the Llama 2 family. Considering the focus on MoE methods, it would be beneficial to include a discussion of findings related to the Mistral model family for a more comprehensive analysis.
- In the abstract, one of the contributions mentioned is the 'optimal merging strategy.' However, to substantiate the claim of an 'optimal strategy,' there needs to be a more detailed explanation of the methods, stronger motivations, and comparisons with more baselines. The reported performance enhancement of 5.61% is compared with the best single model, but comparing a MoE method to a single model may not be entirely fair. Furthermore, based on Table 8, the performance of Model-GLUE does not consistently surpass the full merging method.
- Regarding the selective merging pipeline: (1) the motivation for clustering based on cosine similarity is not clearly articulated; (2) the general criteria for determining the number of clusters should be discussed.
- The concept of a hybrid mixture is intriguing. It would be beneficial to provide a more detailed explanation of the method and the motivations behind it.
- In Section 4.5, Table 2 does not consistently show that the model-level mixture is superior, as suggested in the text. The authors should elaborate further on these results.
- On page 7, Question 3 should reference Table 4 instead of Table 5.
- In general, the captions for tables can be better; some abbreviations such as 'F-L-T' and 'F-L-S' are confusing.
- Q5 on page 8, memory overhead during inference is reduced is not shown in the results.
- In Section 6, where are the experimental results on model stacking?

**Strengths:**

The benchmark results provides by the paper is quite extensive and thorough.

**Additional Feedback:**

N/A.

**Documentation:**

The github page seems to be empty.

**Ethics:**

N/A.

**Limitations:**

Please kindly see above.

**Opportunities For Improvement:**

Please kindly see above.

**Relation To Prior Work:**

The paper would benefit from a more detailed discussion of the contributions and findings based on the benchmarks in the introduction. However, the reported 5.61% improvement of the proposed method is not entirely convincing.

**Summary And Contributions:**

The paper benchmarks existing model merging and mixture-of-experts (MoE) methods using the Llama-2-based model zoo and proposes a comprehensive guideline for scaling large language models (LLMs). It includes extensive experiments conducted across seven tasks. While the paper provides thorough benchmarks comparing existing model merging and MoE baselines, the overall message lacks clarity and could be articulated more effectively.

---

> ### Author Rebuttal · Authors · 2024-08-17
>
> Thanks for acknowledging our experiments and analysis are “extensive and thorough”. We provide pointwise responses to your concerns about our experiment settings below.
>
> **[QUESTION1 Experiment on Mistral model family]**
>
> - Thank you for your suggestion. We choose the Llama2-based model family because there are more diverse variances built on different datasets and training recipes. There are many domain-specific models based on Llama-2, such as those for code[1], mathematics[2], healthcare[3-4], finance[4], law[4], and mental health[5]. Importantly, a series of models have undergone continuous pre-training based on Llama-2 [1-2, 6-8], and a considerable portion of models trained from scratch have drawn inspiration from the architecture of Llama-2 [9-11]. While these models share the same architecture as Llama-2, their weights exhibit significant differences. Thus, we can thoroughly examine the effect of merging, mixture and Model-GLUE on different settings.
> - To further evaluate our proposal on the Mistral model family, we have established a Mistral-based Which8 model zoo and replicated the experiments outlined in Section 5.2. From the result below it can be seen that Mode-GLUE consistently outperform.
>
> Table S1 Comparison between the best single model, Merging, Full Mixture and our Model-GLUE with Mistral model zoo
> |                             | ARC       | WinoGrande | MMLU      | GSM8K | MBPP      | HumanEval | Avg       |
> | --------------------------- | --------- | ---------- | --------- | ----- | --------- | --------- | --------- |
> | Best single model           | **67.24** | 79.01      | 61.77     | 63.15 | 35.98     | 39.00     | 57.69     |
> | DARE Merging [12]              | 64.33     | 78.37      | 63.27     | 63.31 | 39.02     | **44.60** | 58.82     |
> | TIES Merging [13]               | 63.74     | 77.90      | 60.90     | 49.13 | 34.76     | 39.40     | 54.30     |
> | FFN-Linear-Sentence Mixture | 64.85     | **79.72**  | 63.42     | 64.82 | 42.00     | 42.07     | 59.48     |
> | Model-GLUE                  | 65.02     | 78.85      | **64.39** | 65.50 | **44.60** | 42.68     | **60.18** |

---

> > ### Comment · Reviewer_5WA2 · 2024-08-22
> >
> > Thank you for your response. I have 2 quick questions.
> >
> > - Q1: the authors mentioned that the Model-GLUE consistently outperforms, but we can see that it only outperforms in 2 of the tasks. Do authors have any insight/explanations for it?
> > - Q5: how is the threshold 0.95 chosen? Have authors tested out different thresholds and how does it impact the performance?

---

> > > ### Author Rebuttal · Authors · 2024-08-24
> > >
> > > **[Model-GLUE not consistently outperform]**
> > >
> > > Model merging and FFN-level mixture requires models to have the same architecture and initialization. Our Model-GLUE combines progressive merging and model mixture to allow more flexibility. Intuitively, this flexibility would be a tradeoff for model performance. Meanwhile, we surprisingly find that **Model-GLUE can achieve both flexibility and good performance**. As shown in Tables 2 and S1, the average accuracies of Model-GLUE are over the counterparts on both Llama2 and Mistral model families.
> > >
> > > **[Impact of clustering threshold]**
> > > - We computed the cosine similarity between 12 LLMs all fine-tuned from Llama-2. These models are considered to be well mergeable, having the same architecture and initialization. Since their similarities range from 0.9680 to 0.9999, 0.95 could be a lower bound for model clustering.
> > > - To show the impact of different clustering thresholds, we have examined the performance of merged models with drastically different similarity: Llama-2，deepseek-coder, CodeLlama, and MetaMath-Llema. We use linear interpolation to merge two models and present the benchmarking results in Table S2. The performance of the individual models is shown in Table S3. If the merged model outperforms its parent models on average accuracy, we consider it a successful merge.
> > > - From Table S2, we see that successful merging only occurs between Codellama and Codellama-instruct whose weights reach 0.99 similarity and have the same initialization. To include more mergeable models, we finally choose 0.95 as the threshold for clustering.
> > >
> > >
> > >
> > > Table S2 Performance of merged models with different similarity.
> > >
> > > | Recipe Idx | Parent Model 1                     | Parent Model 2                       | ARC    | MMLU   | WinoGrande | GSM8K  | HumanEval | MBPP   | Avg.   | Cosine Similarity | Successful Merge? |
> > > | ---------- | ---------------------------------- | ------------------------------------ | ------ | ------ | ---------- | ------ | --------- | ------ | ------ | ----------------- | ----------------- |
> > > | 1          | meta-llama/Llama-2-7b-hf           | deepseek-ai/deepseek-coder-6.7b-base | 27.73% | 24.38% | 49.64%     | 0.00%  | 0.00%     | 0.00%  | 16.96% | 0.0000             | FALSE             |
> > > | 2          | meta-llama/Llama-2-7b-hf           | codellama/CodeLlama-7b-hf            | 41.04% | 31.68% | 66.85%     | 5.76%  | 10.98%    | 21.40% | 29.62% | 0.5255            | FALSE             |
> > > | 3          | codellama/CodeLlama-7b-Python-hf   | codellama/CodeLlama-7b-hf            | 40.61% | 37.17% | 65.35%     | 6.67%  | 21.95%    | 25.60% | 32.89% | 0.6034            | FALSE             |
> > > | 4          | meta-math/MetaMath-Llemma-7B       | codellama/CodeLlama-7b-hf            | 46.16% | 42.86% | 64.64%     | 27.07% | 34.76%    | 37.40% | 42.15% | 0.8870            | FALSE             |
> > > | 5          | codellama/CodeLlama-7b-Instruct-hf | codellama/CodeLlama-7b-hf            | 43.86% | 41.39% | 68.59%     | 16.07% | 33.54%    | 40.80% | 40.71% | 0.9994            | TRUE              |
> > >
> > > Table S3 Performance of parent models.
> > >
> > > | Models                               | ARC    | MMLU   | WinoGrande | GSM8K  | HumanEval | MBPP   | Avg.   |
> > > | ------------------------------------ | ------ | ------ | ---------- | ------ | --------- | ------ | ------ |
> > > | meta-llama/Llama-2-7b-hf             | 53.92% | 45.83% | 74.11%     | 13.72% | 10.98%    | 18.00% | 36.09% |
> > > | deepseek-ai/deepseek-coder-6.7b-base | 36.86% | 36.36% | 57.30%     | 19.03% | 45.12%    | 54.80% | 41.58% |
> > > | codellama/CodeLlama-7b-hf            | 41.89% | 39.05% | 65.98%     | 11.83% | 32.32%    | 37.20% | 38.05% |
> > > | codellama/CodeLlama-7b-Python-hf     | 40.70% | 35.62% | 64.56%     | 13.12% | 38.41%    | 41.20% | 38.94% |
> > > | meta-math/MetaMath-Llemma-7B         | 46.67% | 46.29% | 64.33%     | 62.24% | 32.32%    | 42.00% | 48.97% |
> > > | codellama/CodeLlama-7b-Instruct-hf   | 43.00% | 41.69% | 65.90%     | 18.12% | 33.70%    | 40.00% | 40.40% |

---

> > > > ### Comment · Reviewer_5WA2 · 2024-08-27
> > > >
> > > > Many thanks for the additional results. I have increased my score to 6.

---

> ### Author Rebuttal · Authors · 2024-08-17
>
> **[QUESTION2 Clarification on the optimal merging strategy]**
>
> - In this study, we consider a more realistic scenario. Given a wild model zoo, there are both useful and harmful models after merging, as shown in Table 10 of Section 4.3. Previous methodologies are fundamentally different from ours in that they define a model zoo and merge them all. This is a subset of our study.  We find models to merge and how to merge them.
> - It is not fair to directly compare our progressive merging to theirs. However, we do experiment with variable control. In Table 12, we compare different merging methods on the same set of models.
> - Our optimal strategy is to select the best methods from heuristic merging, evolutionary merging, and merging methods, as they are the 3 key steps in progressive merging. We will rephrase our terms to reduce confusion.
>
> **[QUESTION3 Fairness of comparing MoE to single model]**
> - Our mixture model and single model have similar activated parameters and efficiency during inference. The overhead for routing networks is minimal. Traditional MoE studies usually compare single dense models [14-17]. In addition, our mixture model does not require training, where model construction adds moderate additional overhead.
> - Thus, it is fair to compare our training-free mixture model to a single model.
>
> **[QUESTION4 Proposed methods not consistently outperform]**
>
> - Our Model-GLUE/model-level mixture differs from its counterparts in settings. Merging and other levels of mixing require the models to have the same architecture and initialization, while our proposals allow more flexibility. The flexibility feature would be an intuitive tradeoff for model performance.
> - Meanwhile, we surprisingly find that Model-GLUE and model-level blending can achieve both flexibility and performance, as shown by their average accuracy in Tables 2 and 8.

---

> ### Author Rebuttal · Authors · 2024-08-17
>
> **[QUESTION5 Details of clustering in selective merging pipeline]**
>
> *Motivation for using cosine similarity as a model selection criterion*
> - Previous merging study [12] finds that merging performance is consistent with parameter similarity. We inherit it by using cosine similarity as a representative method to measure whether a model can be merged.
> - From our preliminary result, cosine similarity works effectively. Empirically, when the cosine similarity between models exceeds 0.95, merging them can yield positive benefits. In Table 10, we present examples of successful and unsuccessful merging. For example, the cosine similarity between the weights of Llama-2-chat and Vicuna is 0.9982, resulting in the merged model significantly outperforming its parent models. On the other hand, the cosine similarity between the weights of Llama-2-chat and CodeLlama is 0.5351, indicating that the merged model is inferior to CodeLlama.
> - Moreover, using cosine similarity to measure the merging benefit is simple and efficient. For these reasons, we stick with cosine similarity for selective merging pipelines.
>
> *Criteria for Determining the Number of Clusters*
> - We cluster models with cosine similarity greater than 0.95 into a mergeable family, ensuring that within this mergeable family, the pairwise similarities between models are greater than 0.95.
> - The number of clusters is automatically determined during the process, after which we execute our merge strategy within each cluster. For "Which16 Model Zoo" in our paper, we clustered 16 models and finally obtained five mergeable families as shown below. Since the remaining clusters contain only one model each, we only report the results of different merging strategies performed within Family 1.
>
> Family 1: 12 models fine-tuned based on llama-2;
>
> Family 2: ise-uiuc/Magicoder-S-CL-7B;
>
> Family 3: codellama/CodeLlama-7b-Instruct-hf;
>
> Family 4: meta-math/MetaMath-Llemma-7B;
>
> Family 5: LLM360/CrystalChat.

---

> ### Author Rebuttal · Authors · 2024-08-17
>
> **[QUESTION6 Explanation of hybrid mixture]**
>
> The implementation details of the hybrid mixture are shown below. We will expand the appendix to include implementation details including the hybrid mixture.
>
> **Hybrid Mixture** derives from the motivation to combine the two effective methods, merging and mixture, for more enhancement. The first k layers of the mixed model are generated by merging manner, and the rest layers are mixed by FFN level mixture manner.
>
> ---
>
> **Algorithm** Hybrid Mixture
>
> ---
>
> **Input:** A model family {$w_1$, ..., $w_n$} with identical layer amount, one of the family as $base_model$, $k$ layers for merging and the rest layers for mixture.
>
> **Output:** $mixed\\_model$
>
> &ensp;1: $mixed\\_model.embedding$ $\longleftarrow$ $base\\_model.embedding$
>
> &ensp;2: $mixed\\_model.lm\_head$ $\longleftarrow$ $base\\_model.lm\\_head$
>
> &ensp;3: **for** $i$ = 0 to $k$ **do**
>
> &ensp;4: $~~~~~$ $mixed\\_model$.$layer_i$ $\longleftarrow$ Merge({$w_1$, ..., $w_n$}, $i$)
>
> &ensp;5: **end for**
>
> &ensp;6: **for** $i$ = $k$ + 1 to LayerNum(base_model) **do**
>
> &ensp;7: $~~~~~$ $mixed\\_model.layer_i.router$ $\longleftarrow$ GenerateRouter($n$)
>
> &ensp;8: $~~~~~$ $mixed\\_model.layer_i.attention$ $\longleftarrow$ $base\\_model.layer_i.attention$
>
> &ensp;9: $~~~~~$ $mixed\\_model.layer_i.norm$ $\longleftarrow$ $base\\_model.layer_i.norm$
>
> 10: $~~~~~$ **for** $j$ = 1 to $n$ **do**
>
> 11: $~~~~~$ $~~~~~$ $mixed\\_model.layer_i.expert_j.FFN$ $\longleftarrow$ $w_j.layer_i.FFN$
>
> 12: $~~~~~$ **end for**
>
> 13: **end for**
>
> 14: **return** $mixed\\_model$
>
> ---
>
> **[QUESTION7 Inappropriate reference on page 7]**
> Thanks for pointing this out. In fact, question 3 should refer to table 5. We will correct this in future versions.
>
> **[QUESTION8 Abbreviation of mixture methods]**
> Thanks for the feedback, we will refine the table captions to use full names for the convenience of the reader in the future paper version.
>
> **[QUESTION9 Inference memory overhead of hybrid mixture]**
> The memory saved by the hybrid mixture primarily resides in the model weights. As the top k transformers blocks in the hybrid mixture are dense, it saves approximately k * (n - 1) Linear layer parameters compared to traditional MoE architectures such as FFN level mixture, where n represents the number of experts.
>
> **[QUESTION10 Missing results of model stacking]**
> We summarize the main results in Section 6 with a reference to Appendix C.2. for the full stacking results. We will include some of them in the main text for completeness.
>
> **[DOCUMENTATION Code issue]**
> Thanks for the detailed review, your question will greatly improve our code management. Our full codes are maintained in [model merging](https://github.com/Model-GLUE/model_merge_llm), [model mixture](https://github.com/Model-GLUE/model_mixture_llm), and [model stacking](https://github.com/Model-GLUE/model_stack_llm), all committed before paper submission. We accidentally put the wrong link in the abstract. We will include them in one repository.

---

> > ### Author Rebuttal · Authors · 2024-08-17
> >
> > **References**
> >
> > [1] Magicoder: Source code is all you need. arXiv:2312.02120.
> >
> > [2] Wizardmath: Empowering mathematical reasoning for large language models via reinforced evol-instruct. arXiv:2308.09583.
> >
> > [3] Meditron-70b: Scaling medical pretraining for large language models. arXiv:2311.16079.
> >
> > [4] Adapting large language models via reading comprehension. arXiv:2309.09530.
> >
> > [5] MentaLLaMA: interpretable mental health analysis on social media with large language models. ACM Web 2024.
> >
> > [6] Code llama: Open foundation models for code. arXiv:2308.12950.
> >
> > [7] Wizardcoder: Empowering code large language models with evol-instruct. arXiv:2306.08568.
> >
> > [8] Metamath: Bootstrap your own mathematical questions for large language models. arXiv:2309.12284.
> >
> > [9] Llm360: Towards fully transparent open-source llms. arXiv:2312.06550.
> >
> > [10] Deepseek llm: Scaling open-source language models with longtermism. arXiv:2401.02954.
> >
> > [11] OpenLLaMA: An Open Reproduction of LLaMA. url: https://github.com/openlm-research/open_llama
> >
> > [12] Language Models are Super Mario: Absorbing Abilities from Homologous Models as a Free Lunch, arXiv:2311.03099.
> >
> > [13] Resolving Interference When Merging Models, NIPS 2023.
> >
> > [14] Switch Transformers: Scaling to Trillion Parameter Models with Simple and Efficient Sparsity, JMLR 2021.
> >
> > [15] Hash Layers For Large Sparse Models, NIPS 2021.
> >
> > [16] ModuleFormer: Modularity Emerges from Mixture-of-Experts, arxiv: 2306.04640.
> >
> > [17] DeepSeekMoE: Towards Ultimate Expert Specialization in Mixture-of-Experts Language Models, ACL 2024.

---

> ### Author Response · Authors · 2024-08-27
> **Thank you for raising the score**
>
> Dear Reviewer 5WA2,
>
> We would like to express our sincere gratitude for your diligent efforts in reviewing our work. We are truly grateful for your valuable time, support, and for raising our score!
>
> Best wishes,
>
> Authors

---

### Official Review · Reviewer_Qy6M · 2024-07-31
**Timely contribution with well-developed framework**

**Rating:** 7
**Confidence:** 2
**Correctness:** Yes
**Clarity:** Yes

**Review:**

**Strength**

The paper is novel, relevant, and timely. The paper is well-developed. Considerations are comprehensive, methodologies are plausible, and the efforts are solid. It holds the potential to be an important contribution to the community.

**Issue**

The text is somewhat dense, but there are still many important details missing.

There are many techniques involved or implemented in the paper, from model merging to mixture models, and so on. However most of the technical procedures or algorithmic implementations are not properly documented. For example, in Section 3, many designs are introduced and compared with where some become an integral part of the proposed framework, yet their technical implementations are entirely missing. In Appendix B, only 3 pseudo-codes are provided with no description of the procedures. This is a major gap in the structure of this paper.

Besides, **The provided code repository is empty.** No usable software is openly provided.

**Strengths:**

See "Review".

**Additional Feedback:**

See "Review".

**Documentation:**

Documentation is incomplete.

**Limitations:**

Recommend discussing the limitations of the proposed framework in a separate subsection or paragraph.

**Opportunities For Improvement:**

See "Review".

**Relation To Prior Work:**

Yes

**Summary And Contributions:**

The paper investigates the research problem of synergetically leveraging a multitude of available LLMs. The paper proposes "Model-GLUE", which aims to provide a holistic guideline for aggregating available LLMs. The paper first benchmarks existing techniques, including selective merging, mixture of experts, etc. The paper then proposes improved, original methods for selecting and aggregating models that can be heterogeneous. The paper conducts empirical validation of the proposed methods with Llama-2 models and achieves performance gain over using standalone models. **The provided code repository is empty.** No usable software is openly provided.

---

> ### Author Rebuttal · Authors · 2024-08-17
>
> We are very glad you had a positive initial impression, and we provide pointwise responses for the concerns of Qy6M as below.
>
> **[QUESTION1 Documentation of technical procedures]**
> We have documented the algorithm implementations of of the most complex three heuristic merging strategies in Appendix B.4, consider the dense information and limited space of the paper main content,
> We will further extend the appendix as shown below, to include three levels of mixture, hybrid mixture and stacking, etc. The evolutionary merging strategies are implemented as documented in the original study [1].
> We will put sufficient details to the final version. If the reviewer has any kind feedback on what additional procedure we need to discuss, we are happy to discuss.
>
> [1] Evolutionary optimization of model merging recipes, arxiv:2403.13187
>
> ### Detailed Algorithms of Model Mixture
>
> **Model Level Mixture.** consists of a router, which determines the expert to execute inference, and all the input models as experts. All the weights of input model's components, including embeddings, layers and LM head, will be integrated into the mixed model.
>
> ---
>
> **Algorithm** Model Level Mixture
>
> ---
>
> **Input:** A model family { $w_1$ , ..., $w_n$ }
>
> **Output:** $mixed\\_model$
>
> &ensp;1: $mixed\\_model.router$ $\longleftarrow$ GenerateRouter($n$)
>
> &ensp;2: **for** $i$ = 1 to $n$ **do**
>
> &ensp;3: $~~~~~$ $mixed\\_model.expert_i$ $\longleftarrow$ $w_i$
>
> &ensp;4: **end for**
>
> &ensp;5: **return** $mixed\\_model$
>
> ---
>
> **Block Level Mixture.** takes an extra base model as input, which is a member of the input model family. The embeddings and LM head of the base model will be taken as corresponding part of the mixed model. Each layer of the mixed model consists of a router and all the corresponding layers of the input models as experts, which will be selected by the router to execute inference.
>
> ---
>
> **Algorithm** Block Level Mixture
>
> ---
>
> **Input:** A model family { $w_1$ , ..., $w_n$ } with identical layer amount, one of the family as $base\\_model$
>
> **Output:** $mixed\\_model$
>
> &ensp;1: $mixed\\_model.embedding$ $\longleftarrow$ $base\\_model.embedding$
>
> &ensp;2: $mixed\\_model.lm\_head$ $\longleftarrow$ $base\\_model.lm\\_head$
>
> &ensp;3: **for** $i$ = 0 to LayerNum(base_model) **do**
>
> &ensp;4: $~~~~~$ $mixed\\_model.layer_i.router$ $\longleftarrow$ GenerateRouter($n$)
>
> &ensp;5: $~~~~~$ **for** $j$ = 1 to $n$ **do**
>
> &ensp;6: $~~~~~$ $~~~~~$ $mixed\\_model.layer_i.expert_j$ $\longleftarrow$ $w_j.layer_i$
>
> &ensp;7: $~~~~~$ **end for**
>
> &ensp;8: **end for**
>
> &ensp;9: **return** $mixed\\_model$
>
> ---
>
> **FFN Level Mixture** is similar to block level with only difference on inner-block component sharing. Each layer of the mixed model will take the attention weights of the base model and build an MoE structure based on the FFNs in corresponding layers of all the input models.
>
> ---
>
> **Algorithm** FFN Level Mixture
>
> ---
>
> **Input:** A model family { $w_1$ , ..., $w_n$ } with identical layer amount, one of the family as $base\\_model$
>
> **Output:** $mixed\\_model$
>
> &ensp;1: $mixed\\_model.embedding$ $\longleftarrow$ $base\\_model.embedding$
>
> &ensp;2: $mixed\\_model.lm\_head$ $\longleftarrow$ $base\\_model.lm\\_head$
>
> &ensp;3: **for** $i$ = 0 to LayerNum(base_model) **do**
>
> &ensp;4: $~~~~~$ $mixed\\_model.layer_i.router$ $\longleftarrow$ GenerateRouter($n$)
>
> &ensp;5: $~~~~~$ $mixed\\_model.layer_i.attention$ $\longleftarrow$ $base\\_model.layer_i.attention$
>
> &ensp;6: $~~~~~$ $mixed\\_model.layer_i.norm$ $\longleftarrow$ $base\\_model.layer_i.norm$
>
> &ensp;7: $~~~~~$ **for** $j$ = 1 to $n$ **do**
>
> &ensp;8: $~~~~~$ $~~~~~$ $mixed\\_model.layer_i.expert_j.FFN$ $\longleftarrow$ $w_j.layer_i.FFN$
>
> &ensp;9: $~~~~~$ **end for**
>
> 10: **end for**
>
> 11: **return** $mixed\\_model$
>
> ---
>
> **Hybrid Mixture** derives from the motivation to combine the two effective methods, merging and mixture, for more enhancement. The first k layers of the mixed model are generated by merging manner, and the rest layers are mixed by FFN level mixture manner.
>
> ---
>
> **Algorithm** Hybrid Mixture
>
> ---
>
> **Input:** A model family {$w_1$, ..., $w_n$} with identical layer amount, one of the family as $base_model$, $k$ layers for merging and the rest layers for mixture.
>
> **Output:** $mixed\\_model$
>
> &ensp;1: $mixed\\_model.embedding$ $\longleftarrow$ $base\\_model.embedding$
>
> &ensp;2: $mixed\\_model.lm\_head$ $\longleftarrow$ $base\\_model.lm\\_head$
>
> &ensp;3: **for** $i$ = 0 to $k$ **do**
>
> &ensp;4: $~~~~~$ $mixed\\_model$.$layer_i$ $\longleftarrow$ Merge({$w_1$, ..., $w_n$}, $i$)
>
> &ensp;5: **end for**
>
> &ensp;6: **for** $i$ = $k$ + 1 to LayerNum(base_model) **do**
>
> &ensp;7: $~~~~~$ $mixed\\_model.layer_i.router$ $\longleftarrow$ GenerateRouter($n$)
>
> &ensp;8: $~~~~~$ $mixed\\_model.layer_i.attention$ $\longleftarrow$ $base\\_model.layer_i.attention$
>
> &ensp;9: $~~~~~$ $mixed\\_model.layer_i.norm$ $\longleftarrow$ $base\\_model.layer_i.norm$
>
> 10: $~~~~~$ **for** $j$ = 1 to $n$ **do**
>
> 11: $~~~~~$ $~~~~~$ $mixed\\_model.layer_i.expert_j.FFN$ $\longleftarrow$ $w_j.layer_i.FFN$
>
> 12: $~~~~~$ **end for**
>
> 13: **end for**
>
> 14: **return** $mixed\\_model$
>
> ---

---

> > ### Author Rebuttal · Authors · 2024-08-17
> >
> > **[QUESTION1 Documentation of technical procedures]**
> >
> > ### Detailed Algorithm of Model Stacking
> >
> > For each input model to stacking, each layer will be absorbed in order into the stacked model. The embeddings and LM heads of the input models will be merged to generate corresponding components of the stacked model.
> >
> > ---
> >
> > **Algorithm** Model Stacking
> >
> > ---
> >
> > **Input:** A model family { $w_1$ , ..., $w_n$ } with identical layer structure
> >
> > **Output:** $stacked\\_model$
> >
> > &ensp;1: $stacked\\_model.embedding$ $\longleftarrow$ MergeEmbeddings({$w_1$, ..., $w_n$})
> >
> > &ensp;2: $stacked\\_model.lm\\_head$ $\longleftarrow$ MergeLMHead({$w_1$, ..., $w_n$})
> >
> > &ensp;3: $index$ $\longleftarrow$ 0
> >
> > &ensp;4: **for** $i$ = 1 to $n$ **do**
> >
> > &ensp;5: $~~~~~$ **for** $j$ = 0 to LayerNum($w_i$) **do**
> >
> > &ensp;6: $~~~~~$ $~~~~~$ $stacked\\_model$.$layer_{index}$ $\longleftarrow$ $w_i$.$layer_j$
> >
> > &ensp;7: $~~~~~$ $~~~~~$ $index$ $\longleftarrow$ $index$ + 1
> >
> > &ensp;8: $~~~~~$ **end for**
> >
> > &ensp;9: **end for**
> >
> > 10: **return** $mixed\\_model$

---

> ### Author Rebuttal · Authors · 2024-08-17
>
> **[QUESTION2 Code issue]**
>
> Thanks for the detailed review, your question will greatly improve our code management. Our full codes are maintained in [model merging](https://github.com/Model-GLUE/model_merge_llm), [model mixture](https://github.com/Model-GLUE/model_mixture_llm), and [model stacking](https://github.com/Model-GLUE/model_stack_llm), all committed before paper submission. We accidentally put the wrong link in the abstract. We will include them in one repository.
>
> **[QUESTION3 Limitation discussion]**
>
> - We place the Discussion of Limitations in Appendix A. We further enrich the content by detailing the limitation situations. The extended version is as follows and will be added to the further paper version:
> - *Our work has the same general limitations as existing studies on LLM scaling. First, a limited theoretical foundation. While empirical evidence suggests that increasing model size, data volume, and computational complexity leads to better performance, there is little theoretical clarity on the exact mechanisms behind these improvements. Second, performance saturation. Although scaling laws suggest that performance continues to improve as models get larger, recent evidence suggests that scaling may lead to diminishing returns beyond a certain point. In addition, our work focuses on benchmarking results, while the reasons for model merging, mixing improves performance could be further improved by post-hoc analysis, such as investigating parameter distribution and similarity during model operations.*

---

### Author Response · Authors · 2024-08-27
**General Response**

We greatly appreciate the time and effort from reviewers and chairs. We are grateful for the valuable comments and suggestions that helped us improve our work. In addition to the point-by-point responses below, we summarize the updates made to our paper.

**[Update 1]** More experiment results on the Mistral model family

To further validate the effectiveness of Model-GLUE as suggested by reviewers **5WA2** and **9iRC**, we set up a Mistral-based model family and implemented model merging, FFN-level mixture, and Model-GLUE. From the benchmarking results, we observed that Mode-GLUE consistently outperforms.

**[Update 2]** More clarification and technical details

- We have documented several algorithm implementations, and model stacking results in the Appendix. We will highlight their references in the main content in the final paper version.
- We extend the appendix to include more technical details, including hybrid mixture, selective merging pipeline, and energy consumption.
- We add more clarification of methods, such as model clustering threshold decision, and consideration for choosing merging baselines.


**[Update 3]** Paper writing revisions

*[Update 3.1]* Update code repository

To address the common concern, we have updated the codebase links for merging, mixture, and stacking, and integrated them into a single repository.

*[Update 3.2]* Title, typos, and references
- Thanks to the careful review of reviewers **5WA2** and **9iRC**, we have fixed some citations and references.
- We will include a more specific limitation discussion as suggested by reviewer **Qy6M**.
- We review the title, method abbreviation, and technical terms. All will be updated in the final version.

Hopefully, our point-by-point responses below can clear up any confusion reviewers may have. If you have any further questions, we would be happy to answer them in detail.

Thanks again for the efforts of all the reviewers and chairs.

---

### Decision · Program_Chairs · 2024-09-26

**Decision:**

Accept (Poster)

**Comment:**

The paper presents a clear and detailed methodology for evaluating different approaches to combining LLMs, providing valuable insights into the strengths and weaknesses of each method. The experiments conducted across multiple tasks demonstrate the effectiveness of certain approaches in improving performance, which can be useful for researchers and practitioners working with LLMs.

Overall, the paper makes a valuable contribution to the field by providing a comprehensive analysis of model merging and MoE methods for scaling LLMs. With some revisions to improve the clarity of the main message and findings, this paper has the potential to become a significant reference for researchers working in this area.